# Variation in temperature of peak trait performance constrains adaptation of arthropod populations to climatic warming

**Samraat Pawar** [1] ✉, **Paul J. Huxley** [1,2] ✉, **Thomas R. C. Smallwood**[3], **Miles L. Nesbit**[1,4], **Alex H. H. Chan** [1], **Marta S. Shocket**[5], **Leah R. Johnson**[2], **Dimitrios - Georgios Kontopoulos** [6] & **Lauren J. Cator** [1] ✉

The capacity of arthropod populations to adapt to long-term climatic warming is currently uncertain. Here we combine theory and extensive data to show that the rate of their thermal adaptation to climatic warming will be constrained in two fundamental ways. First, the rate of thermal adaptation of an arthropod population is predicted to be limited by changes in the temperatures at which the performance of four key life-history traits can peak, in a specific order of declining importance: juvenile development, adult fecundity, juvenile mortality and adult mortality. Second, directional thermal adaptation is constrained due to differences in the temperature of the peak performance of these four traits, with these differences expected to persist because of energetic allocation and life-history trade-offs. We compile a new global dataset of 61 diverse arthropod species which provides strong empirical evidence to support these predictions, demonstrating that contemporary populations have indeed evolved under these constraints. Our results provide a basis for using relatively feasible trait measurements to predict the adaptive capacity of diverse arthropod populations to geographic temperature gradients, as well as ongoing and future climatic warming.

Arthropods are highly diverse and constitute almost half of the biomass of all animals on earth, fulfilling critical roles as prey, predators, decomposers, pollinators, pests and disease vectors in virtually every ecosystem[1]. Arthropod populations are under severe pressure due to pollution and land use changes, which will probably be compounded by ongoing climate change[2–6]. The ability of arthopod populations to adapt to climatic warming in particular has far-reaching implications for ecosystem functioning, agriculture and human health[7–9].

The ability of a population to persist depends on its maximal or 'intrinsic' growth rate ($r_m$) in a given set of conditions[10,11]. The response of arthropod $r_m$ to environmental temperature is unimodal, with its

peak typically occurring at a temperature ($T_{opt}$) closer to the upper, rather than to the lower lethal limit (a left-skewed temperature dependence; Fig. 1a)[12–15]. This temperature dependence of $r_m$ emerges from the thermal performance curves (TPCs) of underlying life-history traits (Fig. 1b)[11,16,17]. Previous work has focused on the effect of thermal sensitivity ('$E$' in Fig. 1b)[11,18] or upper lethal thermal limit ($CT_{max}$)[19–21] of traits on the temperature dependence of $r_m$, providing insights into the responses of populations to short-term thermal fluctuations and heat waves[16,22,23]. However, to understand how populations will respond to long-term sustained climatic warming, we need to quantify the adaptive potential of $T_{opt}$ and the corresponding $r_m$ at that temperature ($r_{opt}$; Fig. 1a).

[1]Department of Life Sciences, Imperial College London, Ascot, UK. [2]Department of Statistics, Virginia Tech, Blacksburg, VA, USA. [3]Department of Infectious Disease Epidemiology, Imperial College London, London, UK. [4]The Pirbright Institute, Woking, UK. [5]Department of Geography, University of Florida, Gainesville, FL, USA. [6]LOEWE Centre for Translational Biodiversity Genomics and Senckenberg Research Institute, Frankfurt, Germany. ✉e-mail: s.pawar@imperial.ac.uk; phuxly@gmail.com; l.cator@imperial.ac.uk

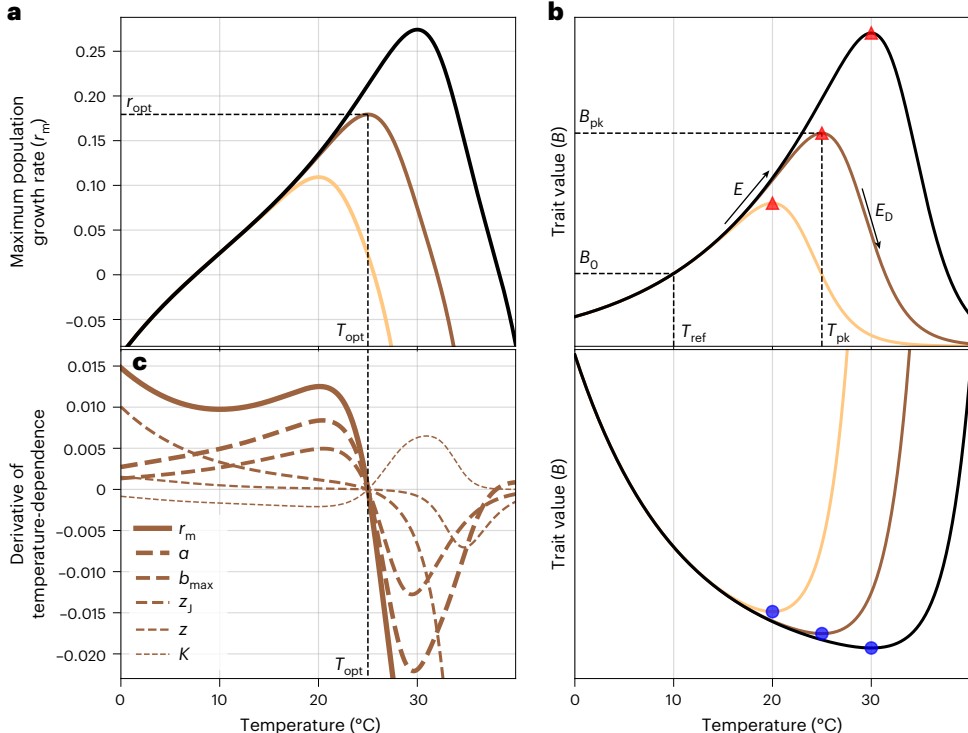

**Fig. 1 | The relationship between the temperature dependence of population fitness and its underlying traits.** In plots **a** and **b** (top and bottom), the three curves (gold, brown and black) represent populations adapted to three different temperatures. **a**, The temperature dependence of population growth rate $r_m$ (equation (2)). Thermal fitness is the peak value that $r_m$ reaches ($r_{opt}$) and $T_{opt}$ is the temperature at which this peak is achieved. **b**, Top: illustration of the underlying trait TPCs modelled using the Sharpe-Schoolfield equation (Methods, equation (3) and Table 1), appropriate for development rate ($1/\alpha$), maximum fecundity ($b_{max}$) and fecundity decline rate ($\kappa$). The red triangles denote the peak trait values ($B_{pk}$) at $T_{pk}$ for each of the populations adapted to three different temperatures. Bottom: illustration of a trait TPC modelled using the inverse of the Sharpe-Schoolfield equation, appropriate for juvenile mortality rate ($z_j$) and adult mortality rate ($z$). The blue circles denote the peak trait values ($B_{pk}$) at $T_{pk}$ for each of the populations adapted to three different temperatures. Both upper and lower sets of curves were generated with arbitrary values for TPC parameters chosen from empirically reasonable ranges for illustration (Methods and Table 1). The $y$-axes values are not shown to keep focus on the qualitative shapes of these TPCs. **c**, Relative contributions of trait TPCs to the temperature dependence of $r_m$. The greater the distance between a trait's partial derivative (dashed curves, that is, $\frac{\partial r_m}{\partial \theta}\frac{d\theta}{dT}$ where $\theta$ denotes any one of the five traits) and the total derivative $\frac{dr_m}{dT}$ (solid curve), the smaller its contribution to $r_m$'s temperature dependence.

In contrast to immediate, rapid or short-term responses to changes in environmental temperatures (that is, phenotypic plasticity), adaptive genotypical shifts in thermal fitness are primarily governed by shifts in the $T_{pk}$s (the temperatures at which trait performance peaks (Fig. 1b); see Table 1 for all parameter definitions) of underlying traits that are captured by changes in $T_{opt}$[11,17]. These trait-specific $T_{pk}$s can evolve relatively rapidly under selection because they are subject to weaker thermodynamic constraints than thermal sensitivity ($E$) or $CT_{max}$[14,24,25]. However, arthropods vary considerably in the form of their complex, stage-specific life histories, and a general mechanistic trait TPC-based approach to quantify their adaptive potential to climate change has proven challenging[9,23,26,27].

Here we combine metabolic and life-history theories to link variation in trait $T_{pk}$s to thermal fitness and adaptive potential in the face of long-term directional changes in temperature (for example, climatic warming) across diverse arthropod populations. Our approach simplifies the diverse complex stage-structures seen in arthropods to the temperature dependence of five life-history traits, allowing general predictions that can be applied across taxa. We test our predictions with a global data synthesis of a diverse set of 61 arthropod taxa to reveal two fundamental constraints on thermal adaptation across arthropod lineages.

## Results

### The trait-driven temperature dependence of fitness

We start with a mathematical equation for the temperature dependence of $r_m$ (Fig. 1a, Methods and equation (2)) as a function of the TPCs

## Table 1 | Definitions of model parameters

| Parameter | Units | Description |
|---|---|---|
| $r_m$ | d⁻¹ | Maximal population growth rate |
| $\alpha$ | d | Egg to adult development time |
| $b_{max}$ | eggs female⁻¹ d⁻¹ | Maximum fecundity rate |
| $\kappa$ | d⁻¹ | Fecundity loss rate |
| $z$ | d⁻¹ | Adult mortality rate |
| $z_j$ | d⁻¹ | Mortality rate averaged across juvenile stages |
| $B$ | measurement unit of trait | Trait value at a given temperature |
| $B_0$ | measurement unit of trait | Normalization constant for trait value at $T_{ref}$ |
| $T_{pk}$ | °C or K | Temperature at which trait performance peaks |
| $B_{pk}$ | Measurement unit of trait | Trait performance achieved at $T_{pk}$ |
| $T_{opt}$ | °C or K | Temperature at which $r_m$ peaks |
| $r_{opt}$ | d⁻¹ | $r_m$ achieved at $T_{opt}$ |
| $E$ | eV | Activation energy, also called thermal sensitivity |
| $E_D$ | eV | Deactivation energy |
| $k$ | eV K⁻¹ | Universal Boltzmann constant (8.617×10⁻⁵) |

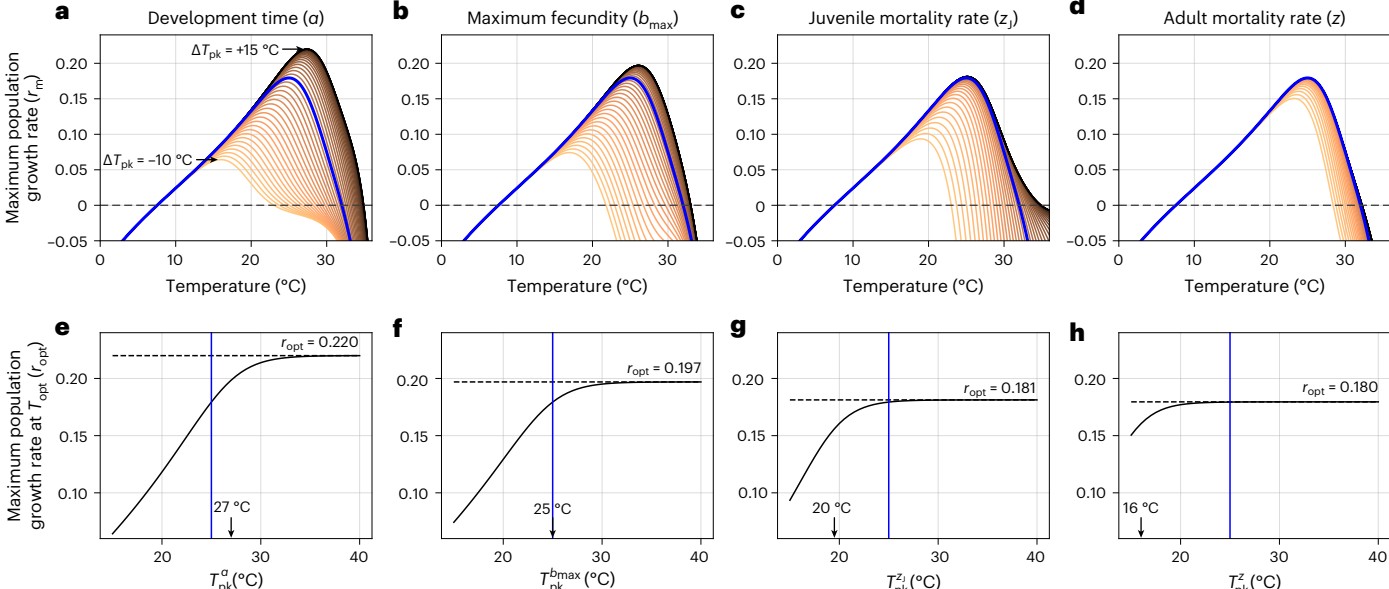

**Fig. 2 | Thermal selection gradients for four key arthropod life-history traits. a–d**, Change in the TPC of $r_m$ with changes in $T_{pk}$s of each of the four traits relative to all the others ($\Delta T_{pk}$). The solid blue line indicates the scenario in which all traits peak at the same temperature ($\Delta T_{pk} = 0$). $\Delta T_{pk}$ ranges from −10 °C (light lines, the focal trait peaks 10 °C cooler than the other traits) to +15 °C (dark lines, the focal trait peaks at 15 °C). **e–h**, The corresponding selection gradients for each trait (Methods). The solid black line represents $r_m$. The solid blue line represents

$\Delta T_{pk} = 0$; any increase or decrease in $r_m$ to the right or left of this point represents potential increase or decrease in thermal fitness that could be gained or lost by increasing or decreasing the $T_{pk}$ of that trait relative to the others. Both sets of plots (**a–d** and **e–h**) are ordered by decreasing strength of selection on the trait, that is, by the temperature at which the selection gradient begins to asymptote (vertical arrows in **e–h**) (Methods).

for five key life-history traits (Fig. 1)[17]: juvenile-to-adult development time, $\alpha$; juvenile mortality rate, $z_j$; maximum fecundity, $b_{max}$; fecundity decline with age, $\kappa$; and adult mortality rate, $z$. This equation predicts that the temperature dependence of $r_m$ increases as a population adapts to warmer temperatures (that is, thermal fitness rises with $T_{opt}$; Fig. 1a). This 'hotter-is-better' pattern is consistent with the findings of empirical studies on arthropods as well as other ectotherms[13,28]. In our theory, this pattern arises through thermodynamic constraints on life-history traits built into the Sharpe-Schoolfield equation for TPCs (Methods and equation (3)), which focuses on a single rate-limiting enzyme underlying metabolic rate[29]. While previous theoretical work has sought to understand the evolutionary basis and consequences of the hotter-is-better phenomenon[13,19,23,25,28,30], to the best of our knowledge, this is presumably the first time that the contributions of underlying traits to it have been quantified. We note that our results do not rely on the specific form or underlying thermodynamic assumptions of the Sharpe-Schoolfield equation; as such, any TPC model that encodes the hotter-is-better pattern, commonly observed across biological traits (Supplementary Results Section 1.1; ref. 11) will yield qualitatively the same results.

**A hierarchy of traits driving thermal fitness.** We next performed a trait sensitivity analysis to dissect how trait TPCs shape the temperature dependence of $r_m$ (Fig. 1c). It shows that populations will grow ($r_m$ will be positive) as long as the negative fitness impact of an increase in juvenile and adult mortality rate ($z_j$ and $z$) with temperature is counteracted by an increase in development and maximum fecundity rate ($\alpha$ and $b_{max}$). So, for example, $r_m$ rises above 0 at −9 °C (Fig. 1a) because at this point $z_j$ and $z$ fall below, and $\alpha$ and $b_{max}$ rise above a particular threshold (Fig. 1c). Similarly, the decline of $r_m$ beyond $T_{opt}$ is determined by how rapidly each of the underlying trait values change with temperature beyond that point (determined by their respective $E_D$s). More crucially, this trait sensitivity analysis reveals a hierarchy in the importance of life-history traits in driving both the location of thermal fitness along

the temperature gradient (that is, $T_{opt}$) and its height (that is, the thermal fitness achieved) (Fig. 1c): the TPC of development time ($\alpha$) has the greatest influence followed by maximum fecundity ($b_{max}$), juvenile mortality ($z_j$), adult mortality ($z$) and fecundity loss rate ($\kappa$), with each of the latter three having a particularly weak effect ~5 °C around $T_{opt}$ (Fig. 1c). It is only at extreme temperatures (Fig. 1c, $T < 10$ °C and $T > 35$ °C) that juvenile mortality ($z_j$) in particular exerts a strong influence on the temperature dependence of $r_m$, consistent with past empirical work[31]. Thus, the influence of the five traits on thermal fitness can be ordered as $\alpha > b_{max} > z_j > z > \kappa$. This hierarchy is expected to be a general result, consistent with the type of life-stage structure typical of arthropod, and especially insect, species where the juvenile stages altogether are both more abundant and longer-lived than adults[16,23].

### The trait hierarchy shapes thermal fitness

Next we focused on trait $T_{pk}$s as key targets of selection leading to thermal adaptation, that is, maximization of thermal fitness in a given constant thermal environment. We calculated thermal selection gradients and strength of selection on each trait's $T_{pk}$ (Fig. 2). For the range of environmental temperatures we consider (0 °C to 35 °C), $T_{opt}$ varies by ~12 °C (−16 °C to 28 °C). Every trait's thermal selection gradient is necessarily asymptotic (Fig. 2e–h) because, even if the $T_{pk}$ of that trait keeps increasing with environmental temperature, all the other traits still decline with temperature beyond their respective $T_{pk}$ (Fig. 1b). These calculations predict that selection is strongest on the temperature of peak performance of development time ($\alpha$), followed by those of the other traits in the same order ($T_{pk}^{\alpha} > T_{pk}^{b_{max}} > T_{pk}^{z_j} > T_{pk}^{z}$), in line with our trait sensitivity analysis (Fig. 1c). We excluded $\kappa$ from this analysis because it has a very weak influence on $r_m$ (Fig. 1c), and TPC data for it are also lacking (Methods). Supplementary Results Section 1.7 shows that our results are indeed robust to variation in $\kappa$'s temperature dependence.

We note that our selection gradient analysis predicts the ordering of strength of selection on traits, ignoring covariances between them.

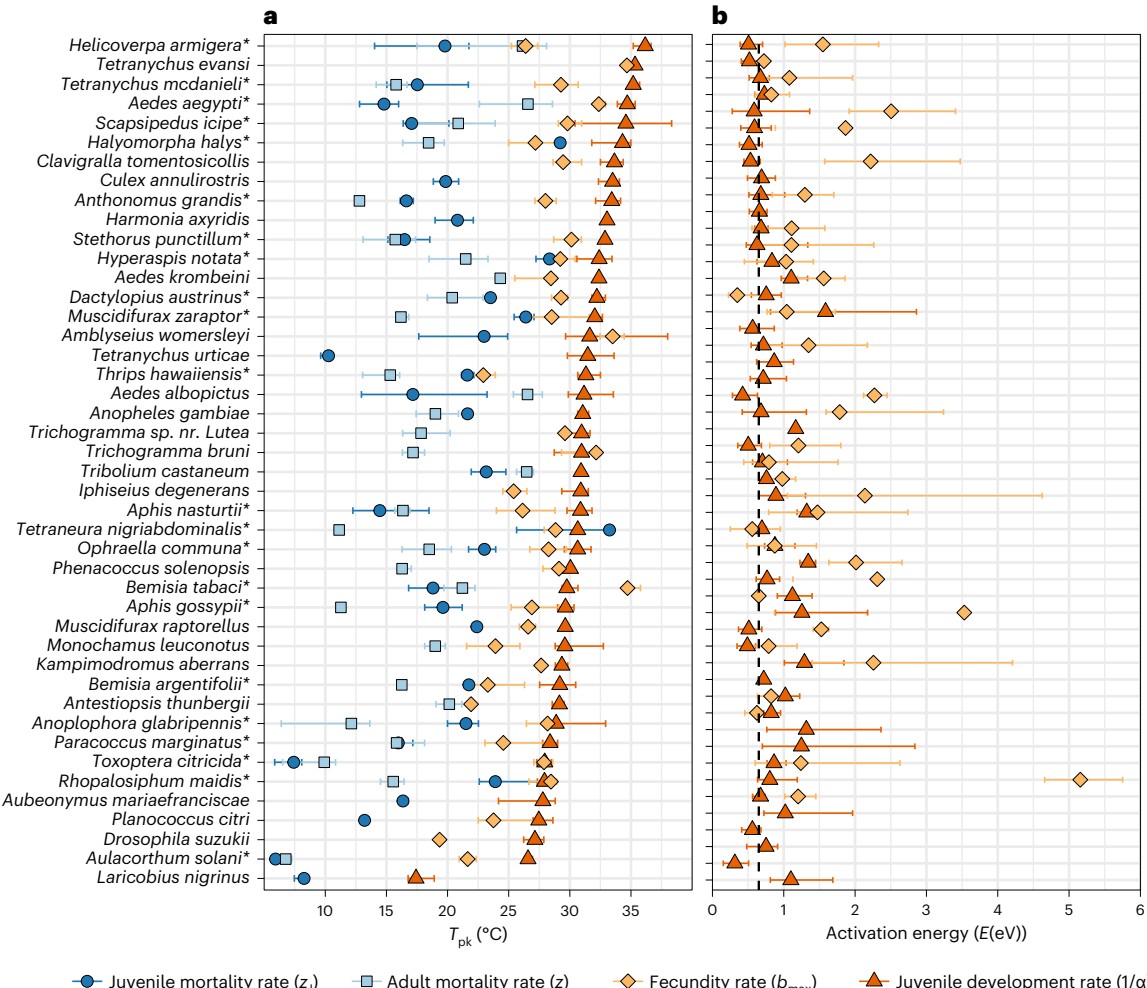

**Fig. 3 | Empirical patterns in arthropod trait peak temperatures and thermal sensitivities. a**, Peak temperatures. **b**, Thermal sensitivities. Each species-(n = 61 examined over 61 independent experiments) and trait-specific thermal response dataset was fitted to the appropriate TPC equation (equation (3) or its inverse; Fig. 1a,b) using NLLS employing the rTPC pipeline[34,64]. Bootstrapping (residual resampling) was used to calculate 95% prediction bounds for each TPC, which also yielded the confidence intervals (error bars) around each median $T_{pk}$ estimate (symbols). Species for whom the full complement of trait data necessary for evaluating the $r_m$ model (Methods, equation (2)) are denoted with asterisks (n = 22 out of n = 44 featured here). Species that did not have data for $\alpha$ (n = 10) or only had $\alpha$ (n = 7) are included in Supplementary Fig. 1.

Covariances between traits, which we address in more detail below, potentially reflect trade-offs between them and would change the shape of the selection gradients (for example, making them unimodal instead of asymptotic) but not the qualitative ordering seen in Fig. 2.

We next tested the theoretical prediction that the trait TPCs underlying thermal fitness evolve hierarchically using an extensive data synthesis that covers 61 different arthropod species (Fig. 3a and Methods). Although this empirical data synthesis largely exhausts the data available in the literature, this relatively small number of species highlights the relative paucity of data on arthropod thermal traits. Nevertheless, our synthesis provides TPC data for an unprecedented diversity of arthropod lineages and allows us to test the generality of our results.

We first tested for the predicted existence of a hierarchical ordering of within-species trait $T_{pk}$s. We found that the empirical patterns are remarkably consistent with the prediction that taxa evolve to optimize their fitness by maximizing the $T_{pk}$ of traits in a specific order (Fig. 2). First, development rate almost always exhibits a higher $T_{pk}$ than peak fecundity, adult mortality rate or juvenile mortality rate (in all but three species; Fig. 3a). Second, in the 22 species for which we were able to find data for all four traits, the ordering of $T_{pk}$s in 55% is exactly as predicted

$(T_{pk}^{\alpha} > T_{pk}^{b_{max}} > T_{pk}^{z_J} > T_{pk}^{z})$. The probability of observing this ordering of trait $T_{pk}$s by random chance is negligible. The match to our theoretical predictions is even stronger if we ignore the data on the trait under weakest selection (adult mortality rate), in which case 68% of the 44 species with data on all three of the more strongly selected traits—development, peak fecundity and juvenile mortality rate—show the expected ordering of $T_{pk}$s. Also, as expected from ecological metabolic theory[11,15,25], the thermal sensitivity parameters (activation energy, $E$) of trait TPCs are relatively constrained across species, emphasizing the primacy of evolution of trait $T_{pk}$s relative to thermal sensitivity in driving thermal adaptation in arthropods (Fig. 3b). Our empirically supported theoretical predictions of a qualitative ordering of the strength of thermal selection on life-history traits are consistent with and reconcile previously scattered empirical results reporting that development rate is more important relative to other traits for ectotherm thermal fitness[31–34], and that it tends to peak at higher temperatures relative to other traits in ectotherms adapting to warmer temperatures[10,35]. Overall, our results provide a key insight: when populations are confronted with long-term (across-generation) climatic warming, we expect the $T_{pk}$s of development rate and the maximum fecundity to shift first and set an upper limit on their thermal adaptation rate.

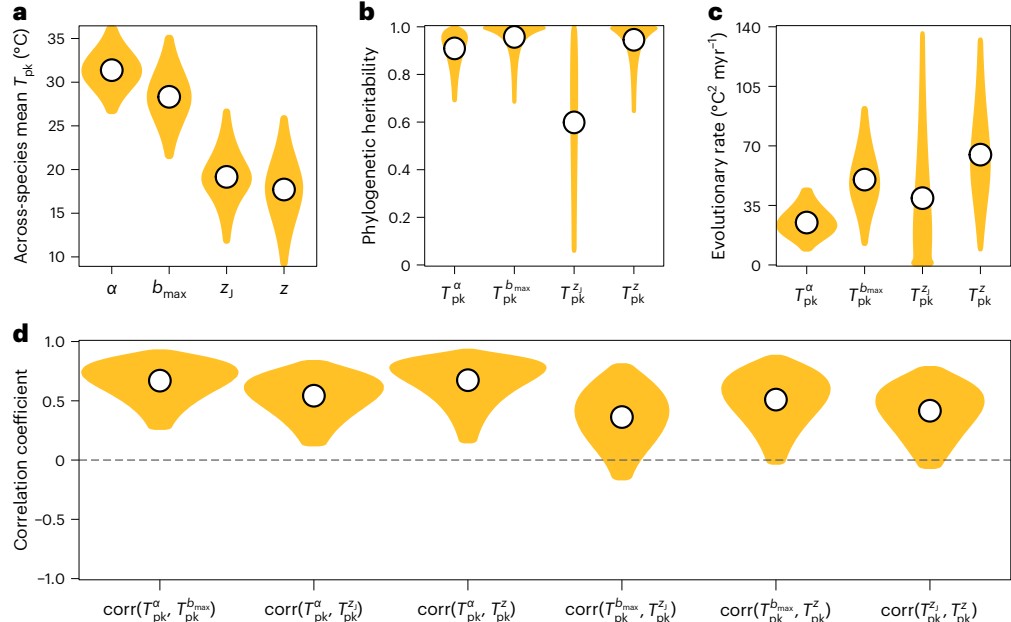

**Fig. 4 | Macroevolutionary patterns of trait $T_{pk}$s across arthropods. a**, The average $T_{pk}$ values across species after accounting for evolutionary relationships. **b**, The fraction of variance in $T_{pk}$ values that is explained by the phylogeny assuming a Brownian motion model of trait evolution (how much more a given trait's $T_{pk}$s for a pair of closely related species are similar than those of randomly selected pairs). The remaining variance arises from phylogenetically independent sources (for example, microevolution and plasticity). **c**, The median rates of evolution (under Brownian motion) per million years. **d**, Median phylogenetic correlations between trait $T_{pk}$ pairs. Gold areas in all plots represent the 95% highest posterior density intervals (Methods).

To investigate the long-term across-species (macro)evolution of trait $T_{pk}$s, we then performed a phylogenetic analysis. The results (Fig. 4a) again support the predicted hierarchy of selection on trait $T_{pk}$s (Fig. 2). The high phylogenetic heritabilities in Fig. 4b imply that adaptive shifts in trait $T_{pk}$s within lineages have not overcome differences across lineages even through long-term evolution. This may be ascribed to systematic differences in metabolic architecture between arthropod lineages[36], but further work linking genomic and thermal metabolic architectures is needed[23]. $T_{pk}^{z_J}$ has the weakest heritability, suggesting a stronger role of plasticity relative to other traits, possibly due to complementary shifts in stage-specific mortality patterns (note that $z_J$ is mortality rate across juvenile life stages). We found weak evidence that $T_{pk}^{\alpha}$ has probably evolved more slowly than $T_{pk}^{b_{max}}$ (Fig. 4c). This is consistent with the strongest stabilizing selection over long macroevolutionary timescales being on $T_{pk}^{\alpha}$. This pattern of stabilizing selection operating over large timescales (millions of years) does not preclude directional, microevolutionary changes in $T_{pk}$s within lineages over shorter few-generation timescales as has been observed (for example, in *Drosophila*[37]). We found that the inferred rate for $T_{pk}^{\alpha}$ is ~5 °C per million years ($\sqrt{\sim 25 °C^2}$; note the *y*-axis scale in Fig. 4c). As such, judging whether this rate is fast or slow needs points of reference and is work for future empirical studies. Here again, we note that these macroevolutionary rates of trait $T_{pk}$s represent averaging across lineages that have experienced thermally changing (for example, high latitudes) as well as stable (for example, tropics) environmental temperatures.

### Physiological mismatches constrain population fitness

As mentioned above, covariances between traits are important for understanding thermal adaptation[14,31,38]. Therefore, we next considered the role of trade-offs and covariances between the $T_{pk}$s of life-history traits in constraining optimization of thermal fitness. Our theory predicts that at a given temperature, the simultaneous maximization of multiple trait $T_{pk}$s should increase thermal fitness exponentially (Fig. 5a). That is, thermal fitness is exponentially higher if the $T_{pk}$s of all traits increase in concert in a warmer environment. For example,

if the development rate $T_{pk}$ increases, the emergent thermal fitness will be higher if the $T_{pk}$s of all other traits increase with it than if they did not. This result, along with our selection gradient analysis (Fig. 2), predicts that selection should favour not just a maximization of $T_{pk}^{\alpha}$ relative to the $T_{pk}$s of other traits, but also a minimization of the differences (that is, 'physiological mismatches') among the trait $T_{pk}$s. However, the constraints of a fixed energy budget would limit such evolutionary optimization, imposing life-history trade-offs. For example, at a given temperature, the available energy can be allocated, for example, to maximize development or fecundity rate, but not both[39,40]. This expectation is supported by our empirical data, which show that there are substantial physiological mismatches in traits across diverse extant arthropod taxa (Fig. 3a), indicating constrained maximization of their $T_{pk}$s. For example, the four species with the highest $T_{pk}$s for development rate (*Helicoverpa armigera*, *Tetranychus mcdanieli*, *Aedes aegypti* and *Scapsipedus icipe*) have as much as a ~20 °C mismatch with the $T_{pk}$s of juvenile mortality and adult mortality rates (which are at the low end of the selection strength hierarchy; Fig. 2).

At the same time, our phylogenetic analysis reveals significant correlated evolution between $T_{pk}^{\alpha}$ and the three other trait $T_{pk}$s (Fig. 4d; median correlation coefficients ~0.5). This is consistent with the expectation that evolution should minimize mismatches among $T_{pk}$s (Fig. 5) to the extent possible despite underlying trade-offs. This positively correlated evolution of trait $T_{pk}$s probably also reflects the fact that despite differences in genomic architecture across arthropods, these four life-history traits share core metabolic pathways, with development in particular most directly linked to fecundity[36,41,42]. For example, in *Drosophila melanogaster*, selection for longer development time has been found to increase early fecundity and decrease late fecundity without notably affecting longevity (hence, mortality rate)[31]. More systematic comparative experimental work is needed to better quantify patterns in the thermal performance of life-history traits across the arthropod tree of life.

It is worth noting that these patterns of correlated evolution of trait $T_{pk}$ are counter to the temperature–size rule[43] that body size decreases with temperature due to accelerated development. On the basis of this

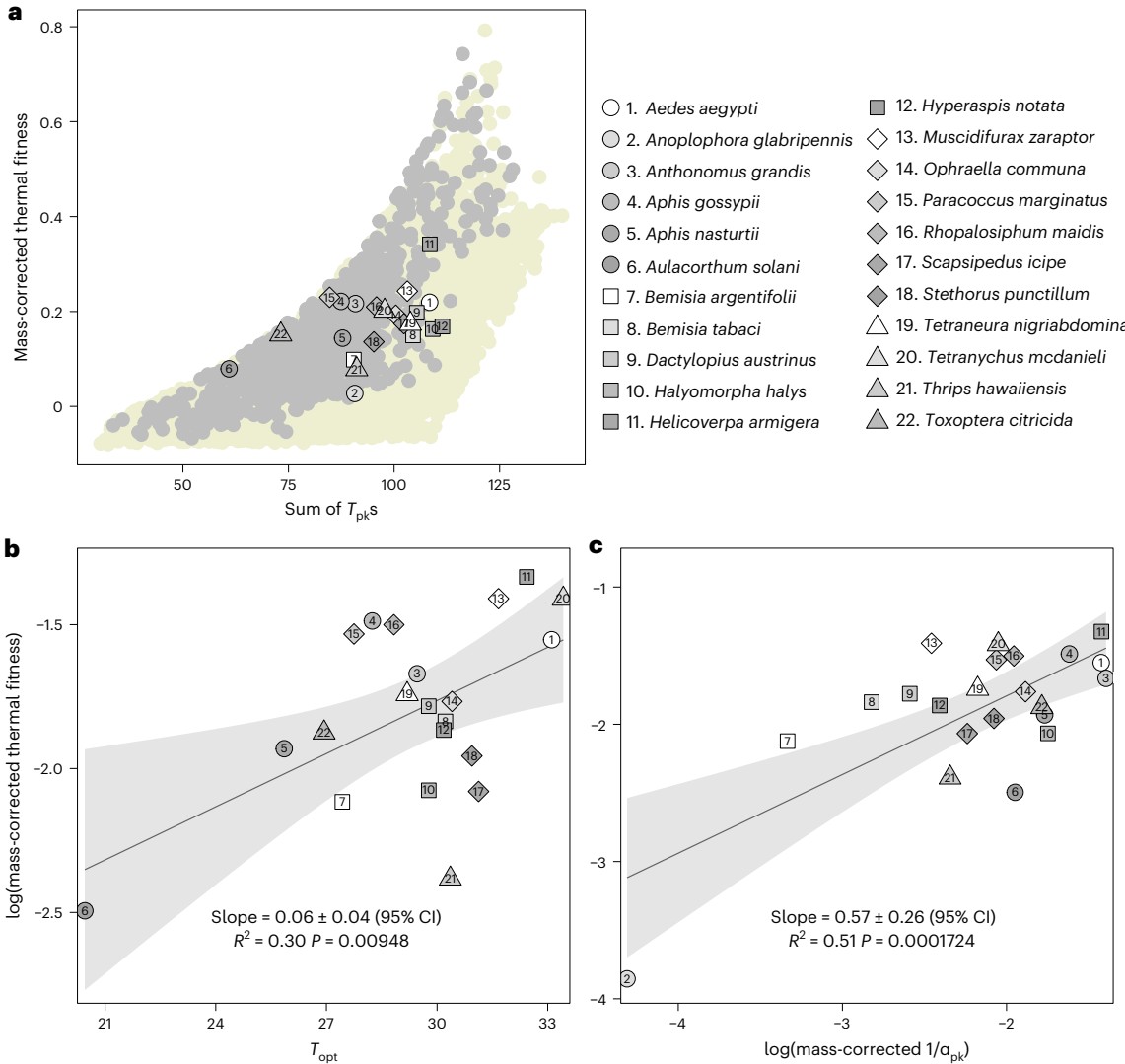

**Fig. 5 | Physiological mismatches predict patterns of thermal fitness in extant arthropod taxa. a**, Thermal fitness ($r_{opt}$) increases with the sum of trait $T_{pk}$s, which quantifies the simultaneous maximization (optimization) of trait TPCs. The pale yellow dots represent simulated thermal fitness obtained by randomly varying the optimal temperatures ($T_{pk}$s) of all traits 5,000 times by drawing them from a uniform distribution between of 10−35 °C (Methods) without constraining the ordering of the trait $T_{pk}$s. Among these, the subset of light grey dots satisfy the predicted hierarchy of trait selection strengths (are closer to an optimal strategy of $T_{pk}^{\alpha} > T_{pk}^{b_{max}} > T_{pk}^{z_j} > T_{pk}^{z}$) (Fig. 2). The thermal fitness values for 22 species (Fig. 3) with sufficient trait data are overlaid on these theoretically predicted strategies. **b,c**, Thermal fitness increases with optimal temperature across a

diverse group of arthropod taxa, that is, a hotter-is-better pattern. **b**, Mass-corrected optimal $r_m$ ($r_{opt}/M^{-0.1}$) predicted by equation (2) increases with its $T_{opt}$. The lines are OLS regression (with 95% prediction bounds) fitted to the species' log-transformed mass-corrected median $r_{opt}$s (symbols) plotted against their respective median $T_{opt}$s. **c**, Consistent with our theoretical prediction, mass-corrected optimal thermal fitness ($r_{opt}/M^{-0.1}$) increases linearly (in log scale) with mass-corrected development rate $T_{pk}$s (($1/\alpha_{pk}$)/$M^{-0.27}$). The lines are OLS regression (with 95% prediction bounds) fitted to the species' log-transformed mass-corrected median $r_{opt}$s (symbols) plotted against their respective mass-corrected median development rate $T_{pk}$s.

rule, we would instead predict a weak positive or even negative correlation between $T_{pk}^{\alpha}$ and $T_{pk}^{b_{max}}$. This is because of a life-history trade-off: on the one hand, a larger body size increases maximal fecundity ($b_{max}$) and reduces mortality rate, thus increasing fitness[44]; on the other hand, growing for a longer duration (higher $\alpha$) increases the mortality realized across juvenile stages and decreases the number of generations completed within a year or season, decreasing fitness. Indeed, many studies have found selection on larval development rate to reduce juvenile mortality and generation time, counterbalancing the positive fitness effect of increasing body size on fecundity[45–47]. This apparent contradiction between the prediction of the size–temperature rule and the macroevolutionary patterns we report here can be reconciled by the fact that the size–temperature rule operates at shorter timescales, potentially affording phenotypic plasticity to populations facing sudden

temperature changes. Future climate change will probably not only entail directional warming but also increases in frequency and magnitude of thermal fluctuations (extreme events). Further theoretical work and empirical data on population fitness ($r_m$) are needed to determine the relative role of phenotypic plasticity versus rapid evolution to build a more complete picture of the ability of arthropod populations to persist under future climate change[19,20,48].

In this context, note also that our TPC data are all from trait measurements at fixed temperatures (Supplementary Results Section 1.3, Source data). This means that our results (Fig. 5) probably overestimate both thermal fitness and its $T_{opt}$ because both tend to decrease under thermal fluctuations[48–50], especially when resource supply is also limiting[33,34]. The two key factors contributing to this phenomenon are physiological stress at high temperatures[49] and fitness benefit

of a larger thermal safety margin (when $T_{opt}$ is relatively lower than median environmental temperature)[50,51] in fluctuating environments. Thus more generally, extending our theoretical framework and thermal fitness calculations to account for patterns of thermal exposure (time-dependent effects[49]) would probably yield important further insights into the constraints imposed by the evolution of trait-specific TPCs (Figs. 1 and 2) on the adaptive potential of arthropod populations.

Finally, to test whether arthropod traits have nevertheless evolved to achieve maximal fitness within the constraints imposed by trade-offs, we overlaid the estimated thermal fitness from the data synthesis onto the theoretically predicted optimal region. We find that real thermal fitness values of diverse arthropod taxa do indeed lie within our theoretically predicted ranges (Fig. 5a). Furthermore, this thermal life-history optimization operates under fundamental thermodynamic constraints, reflected in a global hotter-is-better pattern of thermal fitness (Fig. 5b; note that the 22 species' $r_{opt}$s are within the global hotter-is-better pattern). In addition, as expected, thermal fitness and its hotter-is-better pattern is strongly predicted by the (mass-corrected) value of development rate ($1/\alpha$) at its $T_{pk}$ (Fig. 5c; also see Supplementary Results Section 1.2). We also find that, as expected from our theoretical results, the underlying traits, and most crucially, development time ($\alpha$), also follow a hotter-is-better pattern (Supplementary Results Section 1.1). We also confirmed that these empirical patterns across diverse taxa are the result of long-term adaptive thermal evolution by analysing data on the native thermal environments of these taxa (Supplementary Results Section 1.3).

The above-mentioned constraints are not necessarily the only ones operating. Another possible constraint is the 'jack-of-all-temperatures' pattern (also known as the 'specialist–generalist trade-off'), that is, a negative correlation between performance at $T_{pk}$ and thermal niche breadth[25,52,53]. To get accurate estimates of thermal niche breadth, in particular, numerous trait measurements are required, spanning the entire TPC. In addition, this constraint may apply to some traits but not necessarily all of them, resulting in highly complex patterns. For these reasons, the existence and implications of the 'jack-of-all-temperatures' constraint for arthropod fitness remain to be explored in future studies.

## Conclusions

We have developed a relatively simple theoretical framework to make meaningful predictions about the temperature dependence of population fitness, adaptation potential and climatic vulnerability across diverse arthropods. Crucially, we note that this framework, including the key predictions it makes about the hierarchical importance of life-history traits (Figs. 1c and 2), is general because it captures the 'qualitative' contributions of traits in complex life histories encoded in more complex models[54]. The simplicity of this framework means that it requires data on the TPCs of just five key life-history traits that are in fact relatively measurable across diverse arthropod taxa. This framework also allows constraints imposed by trade-offs between traits' performances, well known to limit fitness optimization in other contexts[55], to be considered in the context of thermal adaptation in a nuanced yet general manner. In particular, our results highlight the fact that the differences in the environmental conditions experienced by juvenile and adult life stages of arthropods warrant particular consideration[23] and may hold the key to developing interventions to counteract the effects of warming on beneficial as well as harmful species. Our overall approach and this framework could thus guide conservation and risk assessment efforts by looking across species to identify those groups most vulnerable to extinction and those most likely to expand their distributions in the face of climate change.

## Methods

### Temperature-dependent maximal population growth rate model

Our model for $r_m$ is based on the Euler–Lotka equation[11,41,44]:

$$\int_\alpha^\infty e^{-r_m x} l_x b_x dx = 1, \tag{1}$$

where $\alpha$ is the age at first reproduction (the time needed for development from egg to adult, or the juvenile-to-adult development time); $l_x$ is age-specific survivorship (proportion of individuals that survive from birth to age $x$); and $b_x$ is the age-specific fecundity (number of offspring produced by an individual female of age $x$). This equation gives the expected reproductive success of a newborn individual in a population growing at a rate $r_m$, under the assumption that the population has reached a stable age distribution (that is, the proportion of individuals in adult and juvenile life stages is constant). Using the simplest feasible mortality and fecundity models (for $l_x$ and $b_x$, respectively), we previously derived an approximation appropriate for the range of growth rates typically seen across arthropods[17] (Table 1).

$$r_m \approx \frac{(\kappa + z)\left(\log\left(\frac{b_{max}}{\kappa + z}\right) - \alpha z_J\right)}{\alpha(\kappa + z) + 1}. \tag{2}$$

Further details of the derivation can be found in Supplementary Information Appendix 2, where we also show that this approximation is very good as long as $r_m$ is relatively small (~<0.5), which is typically where maximal growth rates of arthropods lie[13,17].

Substituting models of the TPCs of the five life-history traits (Supplementary Results Section 1.5) into equation (2) gives the temperature dependence of $r_m$. We model these traits' TPCs using the modified Sharpe-Schoolfield equation (equation (3))[25,29] or its inverse (Fig. 1b):

$$B = B_0 \frac{(E_D - E)e^{\frac{-E}{k}\left(\frac{1}{T} - \frac{1}{T_{ref}}\right)}}{E_D - E + Ee^{\frac{E_D}{k}\left(\frac{1}{T_{pk}} - \frac{1}{T}\right)}}. \tag{3}$$

Here $B$ is the value of a metabolic, ecological or life-history trait at a given temperature ($T$, in K); $B_0$ is a normalization constant representing its value at some biologically meaningful reference temperature ($T_{ref}$); $E$ is the apparent activation energy (initial thermal sensitivity), which determines how fast the curve rises up as the temperature approaches the peak temperature, $T_{pk}$; and $E_D$ is the deactivation energy, which determines how fast the trait declines after the peak. The parameter $k$ is the universal Boltzmann constant ($8.617 \times 10^{-5}$ eV K$^{-1}$). The constant $B_0$ also includes the effect of body size and therefore, the effect of stage-specific size differences, which we do not explicitly consider here theoretically but will consider in the data we analyse below (also see Discussion and Supplementary Results Section 1.1). Equation (3) has been used as a model for thermal performance of traits in numerous previous studies on arthropod population biology because it accurately captures the temperature dependence of a wide range of metabolically constrained life-history traits[15,33,56]. For mortality rates ($z$ and $z_J$), we used the inverse of equation (3) because the thermal response of mortality rate tends to be U-shaped[57–59] and is well-fitted by it[56,60]. Using a different unimodal function instead of the Sharpe-Schoolfield equation does not qualitatively change our results provided that it can encode a 'hotter-is-better' constraint[30]. Substituting the trait TPCs into equation (2) yields the TPC of $r_m$, from which we numerically calculated the $T_{opt}$ (temperature at which the optimum population growth occurs) and $r_{opt}$ (the value of $r_m$ at $T_{opt}$). Note that we refer to the temperature of peak $r_m$ as $T_{opt}$ rather than $T_{pk}$ as we do for traits because the $T_{pk}$s of those traits do not necessarily correspond to the thermal optimum of population fitness (optimal thermal fitness).

### Trait sensitivity analysis

To determine how much influence individual traits' TPCs have on $r_m$'s temperature dependence, using the chain rule we can write[17,57]:

$$\frac{dr_m}{dT} = \frac{\partial r_m}{\partial b_{max}}\frac{db_{max}}{dT} + \frac{\partial r_m}{\partial \alpha}\frac{d\alpha}{dT} + \frac{\partial r_m}{\partial z}\frac{dz}{dT} + \frac{\partial r_m}{\partial z_J}\frac{dz_J}{dT} + \frac{\partial r_m}{\partial \kappa}\frac{d\kappa}{dT}. \quad (4)$$

Each summed term on the right side of this equation quantifies the relative contribution of the TPC of a parameter to the temperature dependence of $r_m$ (Fig. 1c). The trait TPCs for this calculation were parameterized as described above with an identical $T_{pk} = 25\,°C$ across all traits. Here again, the choice of the TPC parameters, as long as all the $B_0$s are in their appropriate scale, does not change the results of this trait sensitivity analysis qualitatively.

### Quantifying trait-specific selection gradients

To quantify the impact of changes in $T_{pk}$s of different traits on $r_m$'s temperature dependence (that is, the thermal selection gradient on each trait), we calculated the shift in optimal maximum population growth rate ($r_{opt}$), that is, the value of $r_m$ at $T_{opt}$ with a unit change in each trait's $T_{pk}$. For this, we re-evaluated equation (2) by varying each trait's $T_{pk}$ in turn while holding all other trait TPCs constant at 25 °C (approximately the median of all the values observed in the data; Fig. 3), over a temperature range of 0–35 °C (Fig. 2). That is, we allowed $T_{pk}$ of each focal trait, in turn, to vary from −10 to +15 °C relative to $T_{pk} = 25\,°C$, keeping all other trait TPCs (including their $T_{pk}$s) fixed. This ensured that the $T_{pk}$s of any given trait varied between 15 °C–40 °C, which is approximately the range seen in the empirical data (see below; Fig. 3). We denoted the difference of the focal trait's $T_{pk}$ from those of all others as $\Delta T_{pk}^{\alpha}$, $\Delta T_{pk}^z$ and so on. The strength of the directional selection gradient for each trait was calculated as the second derivative $\frac{\partial^2 r_{opt}}{\partial T_{pk}^{\theta}}^2$ (where $\theta$ is one of the four traits)[14,24,61,62]. The minimum of this quantity pinpoints the temperature at which that selection gradient starts to asymptote and predicts the ordering of trait $T_{pk}$s that optimizes thermal fitness (Fig. 2). We note that this method ignores trade-offs or covariances between traits, which would change the shape of the selection gradients (for example, making them unimodal instead of asymptotic). We addressed trait covariances through a phylogenetic analysis (below; see the results (Fig. 4) and discussion in the main text).

### Quantifying effects of physiological mismatches

Given that a higher $T_{pk,\alpha}$ increases fitness and that all trait-specific selection gradients increase monotonically, the closer the $T_{pk}$s of the other traits are to $T_{pk,\alpha}$, the higher the $r_{opt}$. We termed the distance between any trait's $T_{pk}$ and $T_{pk,\alpha}$ a 'physiological mismatch'. To quantify the effect of the overall level of physiological mismatch across all pairwise combinations of $T_{pk,\alpha}$ and each of the other three traits, we used the sum of all $T_{pk}$s as a mismatch measure, which is a consistent measure because it increases monotonically with a decrease in mismatch. Then to calculate the effects of this overall physiological mismatch on thermal fitness, we quantified how $r_{opt}$ changes with different potential life-history strategies (in terms of ordering of the $T_{pk}$s) by randomly permuting their order. For this, all other TPC parameters were kept fixed (next).

### TPC parameterizations

For the numerical calculation of $r_m$'s temperature dependence, trait sensitivity, selection gradient analyses and physiological mismatch analyses (above), we parameterized the traits' TPC equations as follows. First, because our theory focuses on the relative differences between the $T_{pk}$s of traits, the exact values of $E$ and $E_D$ do not matter as long as they lie within empirically reasonable values, so we fixed them across all traits to be $E = 0.6$ and $E_D = 4$. These are approximately the median values found in our empirical data (Fig. 3 and Supplementary Fig. 6). Varying these within the range seen in our empirical data do not change our results qualitatively. The parameter values for $B_0$ were varied with type of trait, fixing them to be approximately (rounded off) the median values observed in our empirical data at a $T_{ref}$ of 10 °C: $B_{0,\alpha} = 25$, $B_{0,b_{max}} = 1$, $B_{0,z} = 0.03$ and $B_{0,z_J} = 0.05$ (Supplementary Figs. 7–10). In the absence

of data on the rate of loss of fertility ($\kappa$) or its TPC, we evaluated the model's predictions for a range of values of this parameter (Supplementary Results Section 1.7). While the values for all these parameters in the real data vary considerably around their respective medians, these specific ones used throughout the manuscript's theory figures suffice to generate qualitatively robust theoretical predictions.

### Data synthesis

We performed an extensive literature search to collate existing data on the TPCs of individual traits across arthropod taxa. We searched for publications up to July 2022 using Google Scholar's advanced search, using Boolean operators (for example, life history AND pest OR vector AND temper*) without language restrictions. Additional searches were also made by including species' names in the search string to improve the detection of publications on under-represented groups. For some species, multiple data sets were available, so we only included the study that provided the most complete data (that is, the highest number of traits measured). Raw data and references are available in Source data. We excluded all field studies and also lab studies where traits had been measured over narrow (<10 °C) temperature ranges (that prevented reliable TPC model fitting; below). We also excluded species from our analysis of physiological mismatches if TPC data for at least two traits were not available for them. Values of relevant traits across temperatures were extracted from the text or tables, or were read from the figures using WebPlotDigitizer[63]. All data were converted to consistent measurement units. If juvenile mortality rate ($z_J$, individual$^{-1}$ d$^{-1}$) was not provided (most of the original studies), we divided juvenile survival proportion by development time. Similarly, if adult mortality rate ($z$) was not directly reported, we calculated it as the reciprocal of (typically, female) longevity (d$^{-1}$). When fecundity rate was not directly reported, we divided lifetime reproductive output by longevity to obtain fecundity rate ($b_{max}$; eggs individual$^{-1}$ d$^{-1}$).

Finally, each species- and trait-specific thermal response dataset was fitted to the appropriate TPC equation (equation (3) or its inverse; Fig. 1a,b) using non-linear least squares (NLLS) models employed in the rTPC pipeline[34,64]. Bootstrapping (residual resampling) was used to calculate 95% prediction bounds for each TPC, which also yielded the confidence intervals around each $T_{pk}$ and peak trait value ($B_{pk}$) estimate. Before testing for trait-level hotter-is-better patterns, trait $B_{pk}$s were temperature- and body mass-corrected to account for how optimal thermal fitness ($r_{opt}$) emerges from the TPCs of its underlying traits, which in turn depend on scaling relationships between body size and metabolic rate in individual organisms[44]. Specifically, to obtain the exponents for these relationships, we fitted ordinary least squares (OLS) models in log-log scale (that is, log($B_{pk}$) as a function of log($M$) + $kT$, where $M$ is fresh wet mass in milligrams, $k$ is the Boltzmann constant (Table 1) and $T$ is the trait value at 273.15 K (0 °C)) to the $B_{pk}$ estimates (Main text Fig. 4, and Supplementary Figs. 1 and 2) and the body mass data (Appendix 1). We also corrected $r_{opt}$ to account for size scaling in its underlying traits (Main text Fig. 4 and Supplementary Fig. 2). If fresh mass for a particular species was not provided in the original study, we used mass estimates from other studies on that or a closely related species.

### Phylogenetic analyses

**Phylogeny construction.** To examine the macroevolutionary patterns of the $T_{pk}$s of the four main traits of this study ($\alpha$, $b_{max}$, $z_J$ and $z$), we first extracted the phylogenetic topology of all the species in our dataset from the Open Tree of Life[65] (OTL; v.13.4) using the rotl R package[66] (v.3.0.12). Given that the OTL topology (Supplementary Fig. 16a) included a few polytomies, we also collected publicly available nucleotide sequences (where available) of: (1) the 5′ region of the cytochrome c oxidase subunit I gene (*COI*-5P); (2) the small-subunit ribosomal (r)RNA gene (*SSU*); and (3) the large-subunit rRNA gene (*LSU*). *COI*-5P sequences were obtained from the Barcode of Life Data

System database[67], whereas *SSU* and *LSU* sequences were extracted from the SILVA database[68] (Supplementary Table 1).

Next we aligned the sequences with MAFFT[69] (v.7.490) using the G-INS-i algorithm for *COI*-5P sequences, and the X-INS-i algorithm for *SSU* and *LSU* sequences. We specifically chose the latter algorithm for the two rRNA genes as it can take the secondary structure of RNA into consideration while estimating the alignment[70]. We then removed phylogenetically uninformative sites using the Noisy tool[71] and merged the alignments of the three genes into a single concatenated alignment. For each gene, we identified the optimal model of nucleotide substitution with ModelTest-NG[72,73] (v.0.2.0), according to the small sample size-corrected Akaike Information criterion[74].

For phylogenetic topology inference based on the concatenated alignment, we employed the RAxML-NG tool[75] (v.1.1.0). We constrained the topology search on the basis of the OTL tree, which allowed us to incorporate further phylogenetic information from previously published studies. Nevertheless, because we could not obtain molecular sequences for all species (Supplementary Table 1), some of the polytomies of the OTL tree could not be objectively resolved. To thoroughly account for this uncertainty in downstream analyses, we performed 100 topology searches, each of which started from 100 random and 100 maximum parsimony trees. This process yielded a set of 100 trees, in which 18 alternative topologies were represented (Supplementary Fig. 16b). It is worth pointing out that as few as 50 trees have been found to be typically sufficient for taking phylogenetic uncertainty into account in an analysis[76].

To time-calibrate each of the 100 trees, we first queried the Timetree database[77] to obtain reliable age information (on the basis of at least five studies) for as many nodes as possible. We then applied the 'congruification' approach[78] implemented in the geiger R package[79] (v.2.0.10). In other words, we transferred known node ages (and their uncertainty intervals) from the reference phylogeny (TimeTree) to each of the 100 target phylogenies. This information was then used by the treePL tool[80] to estimate ages for all tree nodes on the basis of penalized likelihood. Supplementary Fig. 16b shows the final set of the 100 alternative time-calibrated trees.

**Investigation of the macroevolutionary patterns of $T_{pk}$s.** To quantitatively characterize the evolution of the $T_{pk}$s of the four traits in this study, we fitted a Bayesian phylogenetic multiresponse regression model with the MCMCglmm R package[81] (v.2.33). In particular, this model had all four $T_{pk}$s as separate response variables with one intercept per response. This allowed us to simultaneously estimate both the variances and covariances among the four $T_{pk}$s and, through this, detect any systematic correlations. Furthermore, we specified a phylogenetic random effect on the intercepts by integrating the phylogenetic variance–covariance matrix into the model. By doing so, we partitioned the variance–covariance matrix of $T_{pk}$s into a phylogenetically heritable component and a residual component. The latter would reflect phylogenetically independent sources of variation in $T_{pk}$ values (for example, plasticity, experimental noise).

Given that we had information on the uncertainty of each $T_{pk}$ estimate from bootstrapping, we directly incorporated this into the model, which makes it a 'meta-analytic' model. We note that this approach effectively ignores covariance between errors, which may have inflated our Type I error. However, satisfactory solutions to this problem are not at present available for the MCMCglmm R package[82]. Missing $T_{pk}$ values for one or more traits per species were modelled as 'missing at random'[83,84]. This approach (not to be confused with 'missing completely at random') allows missing values in a response variable to be estimated (with some degree of uncertainty) from other covarying variables and from the phylogeny, provided that missingness is not systematically driven by a variable that is not included in the model (for example, body size, habitat). Lastly, we specified uninformative priors, namely the default normal prior for the fixed effects, a parameter expanded prior for the random effects covariance matrix and an inverse Gamma prior for the residual covariance matrix.

We fitted this model 100 times, each time with a different phylogenetic tree from our set and with three independent chains per tree. We set the chain length to 200 million generations and recorded posterior samples every 5,000 generations, except for the first 10% of each chain (that is, 20 million generations) which we discarded as burn-in. We then verified that the three chains per tree had statistically indistinguishable posterior distributions and had sufficiently explored the parameter space. For these, we ensured that the potential scale reduction factor value of each parameter was smaller than 1.1 and that its effective sample size was at least equal to 1,000.

Finally, we combined the posterior samples from the three chains of all 100 runs. From these, we first extracted the intercept and the phylogenetically heritable variance of each $T_{pk}$. The former represents the across-species mean, whereas the latter corresponds to the evolutionary rate per million years. We additionally calculated the pairwise correlations between $T_{pk}$s and their phylogenetic heritabilities, that is, the ratio of the phylogenetically heritable variance to the sum of phylogenetically heritable and residual variances. We summarized these parameters by calculating the median value and the 95% highest posterior density interval.

### Reporting summary

Further information on research design is available in the Nature Portfolio Reporting Summary linked to this article.

### Data availability

All data used and generated in the study can be found at https://github.com/EcoEngLab/TraitMismatchPaper-main.git. Our global dataset on arthropods is also available as a Source data file and in the fully open VecTraits database at https://vectorbyte.crc.nd.edu/vectraits-explorer. Source data are provided with this paper.

### Code availability

All code for reproducing the study's analyses can be found at https://github.com/EcoEngLab/TraitMismatchPaper-main.git.

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

## Acknowledgements

We thank R. Huey for insightful comments and suggestions. This work was funded by NIH grant 1R01AI122284-01 and BBSRC grant BB/N013573/1 as part of the joint (NIH-NSF-USDA-BBSRC) Ecology and Evolution of Infectious Diseases programme to L.J.C. and S.P. L.R.J. and P.J.H. were funded by NSF grant DMS/DEB #1750113.

## Author contributions

S.P. and P.J.H. were co-lead authors of the study. S.P. and L.J.C. conceived the study. S.P. developed the theory with support from T.R.C.S., L.R.J. and L.J.C. P.J.H. collated and analysed the data with support from A.H.H.C., M.L.N. and M.S.S. D.-G.K. performed the phylogenetic analyses. All authors contributed to writing and revising the manuscript.

## Competing interests

The authors declare no competing interests.

## Additional information

**Correspondence and requests for materials** should be addressed to Samraat Pawar, Paul J. Huxley or Lauren J. Cator.

# Reporting Summary

## Statistics

For all statistical analyses, confirm that the following items are present in the figure legend, table legend, main text, or Methods section.

| n/a | Confirmed | |
|---|---|---|
| ☐ | ☒ | The exact sample size (*n*) for each experimental group/condition, given as a discrete number and unit of measurement |
| ☐ | ☒ | A statement on whether measurements were taken from distinct samples or whether the same sample was measured repeatedly |
| ☐ | ☒ | The statistical test(s) used AND whether they are one- or two-sided <br> *Only common tests should be described solely by name; describe more complex techniques in the Methods section.* |
| ☐ | ☒ | A description of all covariates tested |
| ☐ | ☒ | A description of any assumptions or corrections, such as tests of normality and adjustment for multiple comparisons |
| ☐ | ☒ | A full description of the statistical parameters including central tendency (e.g. means) or other basic estimates (e.g. regression coefficient) AND variation (e.g. standard deviation) or associated estimates of uncertainty (e.g. confidence intervals) |
| ☐ | ☒ | For null hypothesis testing, the test statistic (e.g. *F*, *t*, *r*) with confidence intervals, effect sizes, degrees of freedom and *P* value noted <br> *Give P values as exact values whenever suitable.* |
| ☐ | ☒ | For Bayesian analysis, information on the choice of priors and Markov chain Monte Carlo settings |
| ☒ | ☐ | For hierarchical and complex designs, identification of the appropriate level for tests and full reporting of outcomes |
| ☒ | ☐ | Estimates of effect sizes (e.g. Cohen's *d*, Pearson's *r*), indicating how they were calculated |

*Our web collection on statistics for biologists contains articles on many of the points above.*

## Software and code

Policy information about availability of computer code

| Data collection | Relevant temperature-trait responses were extracted manually from the text or tables of published literature, or were read from the figures using WebPlotDigitizer (https://automeris.io/WebPlotDigitizer/). |
|---|---|
| Data analysis | All data and code for reproducing the study's analyses can be found at: https://github.com/EcoEngLab/TraitMismatchPaper-main.git. Our global dataset on arthropods is also available as a Source data file and in the fully open VecTraits database (https://vectorbyte.crc.nd.edu/vectraits-explorer). |

For manuscripts utilizing custom algorithms or software that are central to the research but not yet described in published literature, software must be made available to editors and reviewers. We strongly encourage code deposition in a community repository (e.g. GitHub). See the Nature Portfolio guidelines for submitting code & software for further information.

# Data

Policy information about availability of data

All manuscripts must include a data availability statement. This statement should provide the following information, where applicable:

- Accession codes, unique identifiers, or web links for publicly available datasets
- A description of any restrictions on data availability
- For clinical datasets or third party data, please ensure that the statement adheres to our policy

> All data used and generated in the study can be found at: https://github.com/EcoEngLab/TraitMismatchPaper-main.git. Our global dataset on arthropods is also available as a Source data file and in the fully open VecTraits database (https://vectorbyte.crc.nd.edu/vectraits-explorer).

# Research involving human participants, their data, or biological material

Policy information about studies with human participants or human data. See also policy information about sex, gender (identity/presentation), and sexual orientation and race, ethnicity and racism.

| | |
|---|---|
| Reporting on sex and gender | N/A |
| Reporting on race, ethnicity, or other socially relevant groupings | N/A |
| Population characteristics | N/A |
| Recruitment | N/A |
| Ethics oversight | N/A |

Note that full information on the approval of the study protocol must also be provided in the manuscript.

# Field-specific reporting

Please select the one below that is the best fit for your research. If you are not sure, read the appropriate sections before making your selection.

☐ Life sciences ☐ Behavioural & social sciences ☒ Ecological, evolutionary & environmental sciences

For a reference copy of the document with all sections, see nature.com/documents/nr-reporting-summary-flat.pdf

# Ecological, evolutionary & environmental sciences study design

All studies must disclose on these points even when the disclosure is negative.

| | |
|---|---|
| Study description | We performed an extensive literature search to collate existing data on the TPCs of individual traits across arthropod taxa. Each species- and trait-specific thermal response dataset was fitted to the appropriate TPC equation (Eqn 3 or its inverse (Fig. 1A & B) using NLLS using the rTPC pipeline (Padfield 2021; Huxley et al. 2022). Bootstrapping (residual resampling) was used to calculate 95% prediction bounds for each TPC, which also yielded the confidence intervals around each Tpk and peak trait value (Bpk) estimate. These trait value estimates were body mass-corrected to adjust for how the temperature-dependence of rm emerges from the temperature-dependencies of its underlying traits, which, in turn, are dependent on the relationships between body size and metabolic rate in individual organisms (Savage 2004). If fresh mass (mg, M in Main text Fig.4; Supplementary figures 1 & 2) for a particular species was not provided in the original study, we used mass estimates from other studies on that or a closely related species. To test the hotter-better-pattern across taxa, the Bpk values were estimated by fitting either Eqn 3 (for alpha) or its inverse (for zJ and z) for individual species' TPCs using NLLS.<br><br>To examine the macroevolutionary patterns of the Tpks of the four main traits of this study (alpha, bmax, zJ, and z), we first extracted the phylogenetic topology of all the species in our dataset from the Open Tree of Life (OTL; v.13.4) using the rotl R package (v.3.0.12). Given that the OTL topology (Supplementary Fig. 16A) included a few polytomies, we also collected publicly available nucleotide sequences (where available) of: i) the 5' region of the cytochrome c oxidase subunit I gene (COI-5P); ii) the small subunit rRNA gene (SSU); and iii) the large subunit rRNA gene (LSU). COI-5P sequences were obtained from the Barcode of Life Data System database whereas SSU and LSU sequences were extracted from the SILVA database (Supplementary Table 1). |
| Research sample | Our global dataset on arthropods is comprised of mean values for relevant traits across temperatures. These values were extracted from the text or tables, or were read from the figures of published studies using WebPlotDigitizer (Rohatgi 2020). This dataset can be found at: https://github.com/EcoEngLab/TraitMismatchPaper-main.git. It is also available as a Source data file and in the fully open VecTraits database (https://vectorbyte.crc.nd.edu/vectraits-explorer).<br><br>To examine the macroevolutionary patterns of the Tpks of the four main traits of this study (alpha, bmax, zJ, and z), we first extracted the phylogenetic topology of all the species in our dataset from the Open Tree of Life (OTL; v.13.4) using the rotl R package (v.3.0.12). |

| | |
|---|---|
| | Given that the OTL topology (Supplementary Fig. 16A) included a few polytomies, we also collected publicly available nucleotide sequences (where available) of: i) the 5' region of the cytochrome c oxidase subunit I gene (COI-5P); ii) the small subunit rRNA gene (SSU); and iii) the large subunit rRNA gene (LSU). COI-5P sequences were obtained from the Barcode of Life Data System database whereas SSU and LSU sequences were extracted from the SILVA database (Supplementary Table 1). |
| Sampling strategy | Values (means) of relevant traits across temperatures at were extracted from the text or tables, or were read from the figures of published studies using WebPlotDigitizer (Rohatgi 2020). |
| Data collection | PH and MN performed an extensive literature search to collate existing data on the thermal performance curves of individual traits across arthropod taxa. |
| Timing and spatial scale | We searched for publications up to July 2022 using Google Scholar's advanced search, using Boolean operators (e.g., life history AND pest OR vector AND temper*), without language restrictions. Additional searches were also made by including species' names in the search string to improve the detection of publications on under-represented groups. |
| Data exclusions | For some species, multiple data sets were available so we only included the study that provided the most complete data (i.e., the highest number of traits measured). We excluded all field studies, and also lab studies where traits had been measured over narrow (<10 degrees Celsius) temperature ranges (that prevented reliable TPC model fitting). We also excluded species from our analysis of physiological mismatches if TPC data for at least two traits were not available for them. |
| Reproducibility | All data and code used and generated in the study can be found at: https://github.com/EcoEngLab/TraitMismatchPaper-main.git. Our global dataset on arthropods is also available as a Source data file and in the fully open VecTraits database (https://vectorbyte.crc.nd.edu/vectraits-explorer). |
| Randomization | Organisms were allocated to groups based on their scientific species names. |
| Blinding | The objective was to collate all available published data on the temperature-dependence of arthropod fitness traits. Therefore, blinding was not applicable to this study. |

Did the study involve field work?   ☐ Yes   ☒ No

# Reporting for specific materials, systems and methods

We require information from authors about some types of materials, experimental systems and methods used in many studies. Here, indicate whether each material, system or method listed is relevant to your study. If you are not sure if a list item applies to your research, read the appropriate section before selecting a response.

## Materials & experimental systems

| n/a | Involved in the study |
|---|---|
| ☒ | ☐ Antibodies |
| ☒ | ☐ Eukaryotic cell lines |
| ☒ | ☐ Palaeontology and archaeology |
| ☒ | ☐ Animals and other organisms |
| ☒ | ☐ Clinical data |
| ☒ | ☐ Dual use research of concern |
| ☒ | ☐ Plants |

## Methods

| n/a | Involved in the study |
|---|---|
| ☒ | ☐ ChIP-seq |
| ☒ | ☐ Flow cytometry |
| ☒ | ☐ MRI-based neuroimaging |

