## [Peer Review File · Nature Ecology & Evolution]

Peer Review Information

Journal: Nature Ecology & Evolution

Manuscript Title: Variation in temperature of peak trait performance constrains adaptation of arthropod populations to climatic warming

Corresponding author name(s): Samraat Pawar, Paul J. Huxley, Lauren J. Cator

Editorial Notes:

Reviewer Comments & Decisions:

Decision Letter, initial version:
--

13th July 2023

Dear Paul,

I am happy to say that the final review has just come in. Your manuscript has now been seen by 4 reviewers, whose comments are attached below. As you will see from their comments, the reviewers are quite impressed with your work. However, they have also raised some concerns which will need to be addressed before we can offer publication in Nature Ecology & Evolution. We will need to see your responses to the criticisms raised and to some editorial concerns, along with a revised manuscript, before we can reach a final decision regarding publication.

We therefore invite you to revise your manuscript taking into account all reviewer and editor comments. Please highlight all changes in the manuscript text file [OPTIONAL: in Microsoft Word format].

* If you have not done so already please begin to revise your manuscript so that it conforms to our Article format instructions at <http://www.nature.com/natecolevol/info/final-submission>. Refer also to any guidelines provided in this letter.

[REDACTED]

Nature Ecology & Evolution is committed to improving transparency in authorship. As part of our efforts in this direction, we are now requesting that all authors identified as 'corresponding author' on published papers create and link their Open Researcher and Contributor Identifier (ORCID) with their account on the Manuscript Tracking System (MTS), prior to acceptance. ORCID helps the scientific community achieve unambiguous attribution of all scholarly contributions. You can create and link your ORCID from the home page of the MTS by clicking on 'Modify my Springer Nature account'. For more information please visit www.springernature.com/orcid.

[REDACTED]

Reviewer expertise:

Reviewer #1: (insect life history, thermal adaptation)

2Reviewer #2: (physiological ecology, metabolic theory, modelling)

Reviewer #3: (physiological ecology, modelling, scaling)

Reviewer #4: (evolutionary ecology, phylogenetic analysis)

Reviewers' comments:

Reviewer #1 (Remarks to the Author):

This is an interesting and important paper. It examines the thermal sensitivity of a set of life history traits (e.g., development rate, maximum fecundity) and then examines how the temperature dependence of fitness (r_m) is influenced -- in an order-specific way -- by the thermal sensitivity of the underlying traits. That strikes me as insightful. In addition, they examine data on thermal sensitivity of their life history traits for 61 species of insects, and show that the empirical patterns on trait T_{opt} temperatures seem to match expectations. Their model assumes a hotter-is-better pattern, but they also present empirical evidence supporting the hotter-is-better model of the evolution of thermal sensitivity. Finally, in the supplement, they show that trait T_{pk} s are correlated with latitude, which argues that species are weakly adapted with their environments.

That different traits (physiological, life historical) can have different thermal sensitivities has been recognized for decades. Naively, one might expect that the optimal solution would be to have all physiological traits to have congruent TPCs (all have same T_{opt}). But unlike physiological traits (speed, digestion, hearing), the life history traits here have an ontogenetic sequence; and the authors show that the relative importance of these traits varies, with development time having the biggest impacts. That may not be surprising (recall Cole's 1954 result), but I like the way the authors have looked at traits contributing to fitness.

I myself have rudimentary modeling skills and did not attempt to follow all aspects of the model developed here. I'll leave that to other reviewers.

Much as I like this paper, I do need to raise one general issue. All studies that measure the thermal dependence of r_m or T_{pk} have a fundamental problem, namely, that laboratory estimates of the temperature dependence of traits (see fig 3) are derived from fixed temperature experiments (egg to death at fixed temperatures) (see Kingsolver and Woods, 2016, below). But because chronic exposure to high temperature is deleterious (see Kingsolver and Woods, 2016, below), performance or fitness estimates at high temperature will be underestimated. [However, I would be the first to admit that it isn't obvious how one can easily generate a thermal fitness curve without using fixed temperature regimes!] In any case, the authors should at least consider the possible implications of underestimating r_m and T_{pk} at high T_b . **N.B. This concern shouldn't affect their model, but it might affect applications of laboratory derived estimates of T_{pk} (Fig. 3).**

Miscellaneous comments:

38 "the rate of shift in the temperature of peak performance" -- text is unclear 'as is.' I don't know the word limit at Nature for Abstracts, but this Abstract is overly concise (for me).

Miscellaneous comments.

21 the appropriate index for fitness is debated, and r_m is not always preferred (R_0). Check the life history literature (e.g. Kozlowski 1993, Roff 1992, Stearns 1992)

24 recommend: Gilchrist, G. W. (1995). Specialists and generalists in changing environments. I. Fitness landscapes of thermal sensitivity. *American Naturalist*, 146, 252-270.

65 "the TPC of development time (α) has the greatest influence" -- reminiscent of LaMont Cole's 1954 result?

69-70 Impressive result. The importance of development time to r_m has long been appreciated, and was quantified with respect to temperature in Huey and Berrigan 2001. But they examined only development time vs lifetime fecundity (their fig 2).

90 Interesting prediction (Tpk of development rate and max fecundity should shift first), but check <https://doi.org/10.1086/515853> (Gilchrist et al.). Their traits aren't strictly comparable, but they suggest selection may have "had more of an impact on adults than on eggs."

Fig. 2 Very nice. But is it biologically plausible that Tpk of one trait would shift without any change in in Tpk of other traits. That would require that the physiological underpinnings are fully independent.

Is anything known about the heritability and genetic correlations of these traits?

Check papers by Linda Partridge and colleagues on life history responses to laboratory natural selection (temperature).

Partridge et al. (1995) noted: "At the higher environmental temperature, female fecundity was very much higher early in life and declined much more rapidly from the early peak than when measurement was made at the lower environmental temperature."

Check work by Jean David -- I believe he measured the thermal sensitivity of several life history traits.

Fig. 3 Only 1 *Drosophila*? In any case, a very interesting figure. However, did you plot juvenile development TIME or RATE?

107 please expand and add ...parameters (Activation Energies, E)...

118-9 please explain the prediction of stronger stabilizing selection. This came as a surprise, given the importance of development time (90-91). Perhaps I'm confusing something here.

Fig. 5. Suggest that you substitute numbers for the symbols. Thus, change *Bemisia tabaci* to 3, and so on. AS it, a reader will have to work to associate symbols with species. I realize that you are mainly focusing on the overall pattern. Still, I think numbers will help here.

170 I for one would like to see the empirical patterns in the main text rather than buried in the Supplement.

You used a single equation for TPC, but do you suspect your results would hold for the other TPC equations?

183 change "indicate" to "highlight" -- and see Kingsolver and Buckley here.

297 Your way of dealing with polytomies is impressive.

I believe your model does not include tradeoffs for a jack-of-all-temperatures effect (see Huey & Kingsolver, Gilchrist). Should this be mentioned?

In insect studies (e.g., Addo-Bediako et al.), lower lethal temperature drops with latitude much more than upper lethal does. Thus, thermal fitness breadth changes with Tpk. [For latitudinal pattern of Tpk, see Huey 2010.]

s39 Huey (2010, fig 9, citation below) showed that T_{opt} (for r_m) dropped with latitude, but not much!

S41 I'm confused. I assumed the Tpk's were measured from egg to death at fixed temperatures, but here you distinguish between laboratory rearing temperature and rearing temperature. Do you mean stock-culture temperature vs. rearing temperature in a Tpk experiment?

fig. 5 Interesting that "rearing temperature" (I assume this lab-stock temperature) decreases with latitude. Not surprising, but useful to see.

This brings up a point. I haven't checked the original sources for the Tpk data, but are these based on stocks recently from the field or on stocks long adapted to the laboratory? Lab adaptation to temperature is well known and relatively fast in *Drosophila* -- thus 'old' stocks may yield biased results.

If, for example, stocks are adapting to "rearing temperature" (Fig.5) in the lab, then the correlations of trait Tpk with latitude might reflect lab adaptation rather than environmental adaptation.

fig. S7. some of the curves appear to have a single TPC, whereas others have multiple curves (light orange). Please explain. Confidence limits (same issue in fig S9).

s53 Not even in *Drosophila*? There's a substantial aging literature on that group.

####

Some other papers to consider:

Huey, R. B. (2010). Evolutionary physiology of insect thermal adaptation to cold environments. In D. L. Denlinger & R. E. Lee, Jr. (Eds.), *Low Temperature Biology of Insects*. Cambridge University Press.

check a series of papers by Schnebel & Grossfield (mid-1980s) on temperature ranges of mating, oviposition, pupariation etc. in *Drosophila*.

Addo-Bediako, A., Chown, S. L., & Gaston, K. J. (2000). Thermal tolerance, climatic variability and latitude. *Proc. R. Soc. Lond. B*, 267, 739-745. # shows lesser shifts in Tupper-lethal than in Tlower-lethal (or equivalent indices)>

R. B. Huey, J. G. Kingsolver, *Trends Ecol. Evol.* 4, 131 (1989) # graphical depiction of hotter is better and jack-of-all temperatures.

Kingsolver, J. G., & Woods, H. A. (2016). Beyond thermal performance curves: modeling time-dependent effects of thermal stress on ectotherm growth rates. *American Naturalist*, 187, 283-294. <https://doi.org/1086/684786>

Lints, F. A., & Lints, C. V. (1971). Influence of preimaginal environment on fecundity and ageing in *Drosophila melanogaster* hybrids-II. Preimaginal temperature. *Exp. Geront.*, 6, 417-426.

Sinclair, B. J., Marshall, K. E., Sewell, M. A., Levesque, D. L., Willett, C. S., Harley, C. D. G., Marshall, D. J., Helmuth, B. S., & Huey, R. B. (2016). Can we predict ectotherm responses to climate change using thermal performance curves and body temperatures? *Ecology Letters*, 19, 1372-1375. <https://doi.org/10.1111/ele.12686>

Vasseur, D. A., DeLong, J. P., Gilbert, B., Greig, H. S., Harley, C. D. G., McCain, K. S., Savage, V., Tunney, T. D., & O'Connor, M. I. (2014). Increased temperature variation poses a greater risk to species than climate warming. *Proceedings of the Royal Society B*, 281, 20132612.

*** I very much enjoyed reading this manuscript.
Raymond B. Huehy

Reviewer #2 (Remarks to the Author):

Based on the Euler-Lotka equation and the Sharpe-Schoolfield equation, the authors built a model to

6investigate the temperature-dependence of population growth rate, which is a function of five life history traits, each dependent on temperature. The authors performed a series of elegant analyses, which led to two conclusions: the rate of thermal adaptation is limited to the rate of change in temperature of peak performance of four life history traits in a specific order; and thermal adaptation is constrained by tradeoffs between those life history traits.

I read the manuscript several times with great interest. I didn't find any technical flaws in model derivation or statistics. In my opinion, the contribution of this study is very important and very interesting, and the analysis insightful and thought-provoking. Overall, it is THE best manuscript I have reviewed in the last five years. I don't have any major criticism, except for a few questions/suggestions.

1. In the method section, the detailed derivation of Eq. (2), the main equation of the model, is omitted, which is OK, because the interested readers can read the previous paper for the details. But I would suggest that the authors give some intuitive explanations here with words for better understanding.

2. Eq. 3 is equally important. Can the authors give an example to illustrate how body size is included in the constant B_0 ? A related question: I don't quite understand how the authors obtained the scaling powers of body mass to plot Figure 1 and 2 in the supplementary materials.

3. The most important and interesting prediction of this study is the hierarchy of the influence of five traits on thermal adaptation, and therefore, my biggest question is: how do we know that this particular order is not the outcome of this particular model and particular parameter chosen for the analysis (such as Fig. 1-C). How do we know this is a general pattern, instead of model-specific? Yes, the data shows it, but how do we prove its universality theoretically? This is a theoretical question, but I haven't figured out the solution. I would like to hear the authors' insight.

The authors did state: "We note that our results do not rely on the specific form or underlying thermodynamic assumptions of the Sharpe-Schoolfield equation....." But they didn't give detailed explanation or proof, so I am not satisfied.

4. A related question: within the framework of this model, what makes a certain trait, say, development time, more important than the others? Is it due to some specific values of some parameters? Or is it due to their role in Eq. (2)? Or...? I feel that the authors should give some qualitative and intuitive explanations for the biological meanings behind it, (instead of just saying this is the result of our model.

5. A minor suggestion: The legend and the text around Fig. 2 is somewhat hard to read. It took me a while to figure out what the authors were trying to say. Maybe it is just me, but maybe the authors can revise it somewhat?

6. Here is another important question: the only reason given by the authors for the potential trade-offs between the T_{pk} 's of life history traits is the fixed energy budget. As an animal energetics guy, I accept this reason. But I feel this might not be enough; there might be some other reasons. For

7example, the authors mentioned the tradeoffs between development rate and mortality. How would the energy budget argument explain mortality? When it comes to aging, energy budget plays a key role, but when it comes to insects, other external mortality factors, such as predation, instead of intrinsic aging, are more important, so how does energy budget play a role?

Reviewer #3 (Remarks to the Author):

General comments:

For the first time in an impressive, well-thought-out, quantitative analysis, the authors show how the thermal responses of several life-history traits contribute to thermal responses of fitness, as estimated by population growth rate. This analysis has great theoretical and practical value, especially with respect to our understanding of climatic warming on arthropods, a group of animals having great ecological, agricultural, medical and economic importance. I have two general comments that I believe that the authors can easily address.

1) The authors note that their analysis does not consider positive or negative covariation among the life-history traits analyzed (L 242-243). This is understandable given the already complex nature of their analysis. However, trait covariation could alter some of the authors' conclusions, in particular the hierarchy of effects that they have identified on the thermal response of fitness (L 56-72). For example, if developmental time is largely an evolutionary response to juvenile and adult mortality schedules, then saying that developmental time has more of an effect on the thermal response of fitness than do either juvenile or adult mortality may actually not be true. The effects of juvenile & adult mortality may operate indirectly through their effects on developmental time. Whether the authors agree with me or not on this point, I suggest that it might helpful to discuss the above limitation of their analysis in the Discussion section.

2) Are the thermal responses discussed in this paper genotypic evolutionary or phenotypic plastic responses – i.e., 'thermal adaptation' or thermal acclimation' (or both)? The term "thermal adaptation" is frequently used in this manuscript, and the authors focus on temperature effects on evolutionary fitness, so it seems that the authors are talking about evolutionary responses. Please clarify. The answer to this question bears on my 1st comment above.

Specific comments:

L 30: What is a Tpk? Please define at first use. How is it different from Topt, which seems to be the same? Reference here to Fig. 1B is not completely helpful in this respect -- it would be helpful to the reader to explain the difference in words in the text (at first use, and not until L 218-220). A reference to the glossary in Table 1 in the Methods would also be helpful.

L 153-154: I agree that the relative effects of genotypic evolution and phenotypic plasticity on thermal responses (including the temperature-size rule) should be distinguished. Many investigators have studied the T-size rule with acute T changes in the laboratory. A common explanation for the T-size

8rule in these cases is that maturation rate is more T-sensitive than growth rate. The authors discuss how evolutionary thermal responses may not follow the T-size rule, but they have not estimated thermal responses of growth rate, which are critical for understanding this rule.

L 200: Change "traits" to "trait"?

Reviewer #4 (Remarks to the Author):

I was asked to look at the part of phylogenetic comparative analyses (PCA) in this MS. So I am just going to do that although I did scan the other parts and find them very interesting. I believe the part for PCAs were done well and competently. I have just a minor comments.

1. it says "a Cauchy prior for the random effects covariance matrix,". The parameter expanded prior (called by Hadfield) is a prior using the non-central F distribution so it is not quite a Cauchy although a special case of it is a half Cauchy. So the authors may call it just the parameter expanded prior.

2. I checked the code provided and it is very nice - it could do with more annotation (but it is fine). Actually, they use a "meta-analytic" model incorporating SE^2 (using the mev argument). This could be mentioned in the method section. Actually, in a meta-analytic literature, they recommend modelling covariance between errors (i.e SE^2). Otherwise, you may have more Type 1 error but there is no easy solutions for this (at least one cannot do it with MCMCglmm). For example, see

Mavridis D, Salanti G. A practical introduction to multivariate meta-analysis. *Statistical methods in medical research*. 2013 Apr;22(2):133-58.

3. the authors do not justify why they used 100 trees. Actually, this paper shows 50 trees are enough to correct for phylogenetic uncertainty - so 100 is a good one.

Nakagawa S, De Villemereuil P. A general method for simultaneously accounting for phylogenetic and species sampling uncertainty via Rubin's rules in comparative analysis. *Systematic Biology*. 2019 Jul 1;68(4):632-41.

*****END*****

Author Rebuttal to Initial comments

Responses to the Reviewers

Comments made by Reviewers are in **blue text**, whereas our responses are in **black text**.

All revisions to the manuscript referenced below are in **red text** in the revised manuscript.

Reviewer #1

1. All studies that measure the thermal dependence of r_m or T_{pk} have a fundamental problem, namely, that laboratory estimates of the temperature dependence of traits (see fig 3) are derived from fixed temperature experiments (egg to death at fixed temperatures) (see Kingsolver and Woods, 2016, below). But because chronic exposure to high temperature is deleterious (see Kingsolver and Woods, 2016, below), performance or fitness estimates at high temperature will be underestimated. [However, I would be the first to admit that it isn't obvious how one can easily generate a thermal fitness curve without using fixed temperature regimes!] In any case, the authors should at least consider the possible implications of underestimating r_m and T_{pk} at high T_b .

We thank the Reviewer for raising the very crucial issue of the effects of duration and pattern of thermal exposure (time-dependent effects). However, we believe the Reviewer means “overestimating” rather than “underestimating”, as ramping or fluctuating temperatures would both further depress fitness and decrease the thermal optimum^{1,2} especially when resource supply is also limiting^{3,4}, which might be partly compensated by acclimation (e.g., through heat shock proteins) under chronic sub-lethal exposure. While we had previously addressed the effects of thermal fluctuations, we now address time-dependent effects more generally and directly in lines 176-185 of the revised manuscript.

2. line 8: “the rate of shift in the temperature of peak performance” – text is unclear ‘as is’. I don't know the word limit at Nature for Abstracts, but this Abstract is overly concise (for me).

The reviewer is right - our abstract was unnecessarily short by 50 words! We have now revised it, and in the process have clarified this sentence.

3. line 21: the appropriate index for fitness is debated, and r_m is not always preferred (R_0). Check the life history literature (e.g. Kozlowski 1993, Roff 1992, Stearns 1992)

We are aware of this debate / uncertainty and therefore used the phrase “depends on its maximal...” in our original manuscript. As such, we believe this is sufficient, and therefore would rather not qualify the statement further. Nevertheless, we have added two references to direct the reader towards the appropriate literature (line 24).

10

4. line 24 recommend: Gilchrist, G. W. (1995). Specialists and generalists in changing environments. I. Fitness landscapes of thermal sensitivity. *American Naturalist*, 146, 252-270.

We agree and now cite this work (now line 26).

5. line 65: “the TPC of development time (α) has the greatest influence” – reminiscent of LaMont Cole’s 1954 result?

We assume the reviewer is referring to Cole (1954; DOI:10.1086/400074). Cole does not consider temperature but does find that development rate is a very important life history trait in general, as it prolongs the onset of reproduction. We now cite this paper on line 96 to give context to the history of previous work that has highlighted the importance of development rate.

6. lines 69-70 : Impressive result. The importance of development time to r_m has long been appreciated, and was quantified with respect to temperature in Huey and Berrigan 2001. But they examined only development time vs lifetime fecundity (their fig 2).

We now also cite this paper on lines 24 and 97 to give context to the history of previous work on the importance of development rate.

7. line 90: Interesting prediction (T_{pk} s of development rate and max fecundity should shift first), but check <https://doi.org/10.1086/515853> (Gilchrist et al.). Their traits aren't strictly comparable, but they suggest selection may have “had more of an impact on adults than on eggs.”

We thank the Reviewer for drawing our attention to this study. However, Gilchrist et al did not systematically measure shifts in the T_{pk} s (instead focused on heat and cold tolerance) of the 4 traits we refer to in our prediction, so their qualitative result is hard to directly relate to it. Instead, we now cite this paper in the context of the importance of the shape of TPCs (line 26) and correlations between thermal traits, both of which which Gilchrist et al emphasise (line 136).

8. Fig. 2 Very nice. But is it biologically plausible that T_{pk} of one trait would shift without any change in T_{pk} of other traits. That would require that the physiological underpinnings are fully independent.

Agreed. The selection gradient analysis was meant to predict the *qualitative* ordering of the strength of selection on traits *ignoring the* correlations between them. We then explored the correlations between traits (potentially reflecting trade-offs) through the phylogenetic analyses. We had previously provided a caveat about this in the Methods, which we have now moved to the main text, where we also now give an indication to the reader that an examination of the correlations between traits will come later in the paper (lines 90-93). This discussion of correlations and trade-offs between traits can be found in lines 134-145 and 152-161).

11

9. Is anything known about the heritability and genetic correlations of these traits? Check papers by Linda Partridge and colleagues on life history responses to laboratory natural selection (temperature).

Partridge et al. (1995) noted: "At the higher environmental temperature, female fecundity was very much higher early in life and declined much more rapidly from the early peak than when measurement was made at the lower environmental temperature."

Check work by Jean David – I believe he measured the thermal sensitivity of several life history traits.

Fig. 3 Only 1 *Drosophila*? In any case, a very interesting figure.

s53 Not even in *Drosophila*? There's a substantial aging literature on that group.

Again, we thank the Reviewer for bringing up the key issue about trait / genetic correlations and providing these references (also see response to comment 8). We now address this issue and cite a recent review⁵ focusing on *Drosophila melanogaster* in lines 156-161 of the main text. In that taxon, there is considerable work on correlations (including genetic) and apparent trade-offs between arthropod life-history traits. However, the results are somewhat ambiguous, and very little is known about the underlying mechanisms / physiological underpinnings⁵. This is an important unresolved problem that limits our understanding of constraints on the evolution of arthropod life histories across temperatures. Indeed, Flatt et al⁵ show that this has not been worked out even in *Drosophila melanogaster*, arguably the arthropod taxon where thermal adaptation has been studied the most.

Regarding the availability of multiple trait TPC data in *Drosophila*: Very few papers have measured more than one trait at sufficient temperature points over a broad enough range to allow the inference of TPC parameters⁶.

Regarding the rate of decline in fecundity with time (the κ parameter): We have only found *one* study⁷ that reports the data (see their Fig. 2) needed to infer the TPC of κ across *all* arthropods. In any case, our results show that this parameter has weak a relatively impact on thermal adaptation (SI Section 1.7). If the Reviewer is aware of some studies that can help fill this data gap, we would be very excited!

10. However, did you plot juvenile development TIME or RATE?

To estimate T_{pk} for development, we fitted the Sharpe-Schoolfield equation to the inverse of development time (i.e., development rate; $1/\alpha$). We did this because it is well-established that this trait responds to temperature in a unimodal negatively skewed way⁸. This was explained in the Methods, but we have now also made it clearer by changing the label from α to $1/\alpha$ in in Fig. 3. We made this change to Fig. 3 because we plotted the T_{pk} estimates from the trait TPC fits (Supplementary figures 7 - 10). However, we use α_{pk} elsewhere because development rate was inverted to give α for the r_m calculations.

12

11. 107 please expand and add ...parameters (Activation Energies, E)...

Done (lines 116-117).

12. 118-9 please explain the prediction of stronger stabilizing selection. This came as a surprise, given the importance of development time (90-91). Perhaps I'm confusing something here.

Under long-term warming, there will be directional selection which should be strongest for development rate. However, across the timescales of the phylogeny (millions of years), there hasn't been a general trend of long-term warming. So, at a very coarse phylogenetic scale (across all lineages in the tree), the T_{pk} of development rate would not be expected to evolve directionally. Our finding of a lower rate of evolution for T_{pk} of development rate than that of maximum fecundity therefore suggests that across the entire phylogeny (so over macroevolutionary time), this is due to stabilising selection. It is indeed likely that individual lineages have experienced directional selection due to chronic cooling or warming of their environment, but a general pattern of directional selection would not emerge from this at the level of the phylogeny. We now clarify this in lines 124-132 of the main text.

13. Fig. 5. Suggest that you substitute numbers for the symbols. Thus, change Bemisia tabaci to 3, and so on. AS is, a reader will have to work to associate symbols with species. I realize that you are mainly focusing on the overall pattern. Still, I think numbers will help here.

We have added numbers to main Fig 5 and SM Fig 2. Thank you for this suggestion.

14. 170 I for one would like to see the empirical patterns in the main text rather than buried in the Supplement.

We agree with the Reviewer, but Nature E&E's figure number constraints force us to keep this in the SI.

15. 183 change "indicate" to "highlight" – and see Kingsolver and Buckley here.

Done (line 214).

16. 297 Your way of dealing with polytomies is impressive.

I believe your model does not include tradeoffs for a jack-of-all-temperatures effect (see Huey & Kingsolver, Gilchrist). Should this be mentioned?

In insect studies (e.g., Addo-Bediako et al.), lower lethal temperature drops with latitude much more than upper lethal does. Thus, thermal fitness breadth changes with T_{pk} . [For latitudinal pattern of T_{pk} , see Huey 2010.]

This is a great point. However, to test this, we would need to test for the covariance between maximum performance and niche width. The problem is that the jack-of-all-temperatures

13

pattern may apply to some traits but not all. Therefore, we have acknowledged this as a potential limitation and future direction (lines 198-204).

17. s39 Huey (2010, fig 9, citation below) showed that T_{opt} (for r_m) dropped with latitude, but not much!

In fact, while a clear signature of thermal adaptation across geographic gradients would be expected, real-world data include several other factors that influence adaptation. For example, we have also found weak evidence in plants too⁹. We now address this in the SM citing Huey (2010)¹⁰(SM lines 40-50).

18. S41 I'm confused. I assumed the T_{pk} were measured from egg to death at fixed temperatures, but here you distinguish between laboratory rearing temperature and rearing temperature. Do you mean stock-culture temperature vs. rearing temperature in a T_{pk} experiment?

We use 'rearing temperature' to refer to the constant temperature at which a species was reared in the laboratory prior to its use in an original study. We have now made this clearer on SM lines 45-46.

19. fig. 5 Interesting that "rearing temperature" (I assume this lab-stock temperature) decreases with latitude. Not surprising, but useful to see.

This brings up a point. I haven't checked the original sources for the T_{pk} data, but are these based on stocks recently from the field or on stocks long adapted to the laboratory? Lab adaptation to temperature is well known and relatively fast in *Drosophila* – thus 'old' stocks may yield biased results.

If, for example, stocks are adapting to "rearing temperature" (Fig.5) in the lab, then the correlations of trait T_{pk} with latitude might reflect lab adaptation rather than environmental adaptation.

Regarding our terminology for this, please see previous comment above.

Regarding the signature or legacy of the lab/rearing temperature environment/history on single species' trait TPCs: We agree this is an interesting and important question. The populations used in our study probably vary considerably in the number of generations they have been reared in the lab. As the reviewer says, if a particular species' population has been reared in the lab for a long enough time, any correlation between trait T_{pk} and lab temperature would need to be interpreted carefully. This is the reason why we present SM Fig. 5: we were searching for evidence that experimentalists tend to rear their critters at temperatures closest to that species' population's native range / location and as such SM Fig. 5 reveals only weak evidence for this. This, combined with the fact that SM Fig. 3-4 show clear (albeit weak) evidence of the expected relationship between latitude or rearing temperature and T_{pk} , and our phylogenetic analyzes lead us to conclude that

yes, there is sufficient evidence of systematic thermal adaptation of trait TPCs in our collated dataset.

There is also the question of the rate of adaptation: how many generations does it take to completely erase the signature of thermal adaptation of the ancestral population? The jury is still out there on this, or is yet to be assembled (apologies for the metaphor - Trump's Georgia court case weighs on our minds); that is, data on this are largely missing or are yet to be synthesised^{11,12}. In particular, Hoffman et al.¹² found that arthropod life history traits can adapt quickly to laboratory conditions, but their review only focused on (or was able to address) a few temperature-related effects: how lab-rearing can affect tolerance to extreme temperatures.

We now address this issue in SM lines 40-50.

20. fig. S7. some of the curves appear to have a single TPC, whereas others have multiple curves (light orange). Please explain. Confidence limits (same issue in fig S9).

The reviewer is referring to the prediction bounds in some cases suggesting that there may be more than one peak (e.g., *Laricobius nigrinus* and *Monochamus leuconotus*). This is due to some temperatures having single data points so reflects statistical / data uncertainty, not necessarily multiple actual peaks. This is because we use a robust method, bootstrapping, to estimate confidence bounds around each fitted / estimated TPC¹³. Specifically, this uncertainty arises from fitting the TPC model (in our case, the Sharpe-Schoolfield equation) to TPC data where the peak is only captured through a single trait measurement at at or close to that temperature. As such, this does not affect our results qualitatively. We now clarify this in the text legend of the the SM Fig. 7.

21. Some other papers to consider:

Huey, R. B. (2010). Evolutionary physiology of insect thermal adaptation to cold environments. In D. L. Denlinger & R. E. Lee, Jr. (Eds.), *Low Temperature Biology of Insects*. Cambridge University Press.

check a series of papers by Schnebel & Grossfield (mid-1980s) on temperature ranges of mating, oviposition, pupariation etc. in *Drosophila*.

Addo-Bediako, A., Chown, S. L., & Gaston, K. J. (2000). Thermal tolerance, climatic variability and latitude. *Proc. R. Soc. Lond. B*, 267, 739-745. - shows lesser shifts in Tupper-lethal than in Tlower-lethal (or equivalent indices)

R. B. Huey, J. G. Kingsolver, *Trends Ecol. Evol.* 4, 131 (1989) graphical depiction of hotter is better and jack-of-all temperatures.

Kingsolver, J. G., & Woods, H. A. (2016). Beyond thermal performance curves: modeling time-dependent effects of thermal stress on ectotherm growth rates. *American Naturalist*, 187, 283-294. <https://doi.org/1086/684786>

Lints, F. A., & Lints, C. V. (1971). Influence of preimaginal environment on fecundity and

ageing in *Drosophila melanogaster* hybrids-II. Preimaginal temperature. *Exp. Geront.*, 6, 417-426.

Sinclair, B. J., Marshall, K. E., Sewell, M. A., Levesque, D. L., Willett, C. S., Harley, C. D. G., Marshall, D. J., Helmuth, B. S., & Huey, R. B. (2016). Can we predict ectotherm responses to climate change using thermal performance curves and body temperatures? *Ecology Letters*, 19, 1372-1375. <https://doi.org/10.1111/ele.12686>

Vasseur, D. A., DeLong, J. P., Gilbert, B., Greig, H. S., Harley, C. D. G., McCain, K. S., Savage, V., Tunney, T. D., & O'Connor, M. I. (2014). Increased temperature variation poses a greater risk to species than climate warming. *Proceedings of the Royal Society B*, 281, 20132612.

We thank the reviewer for these suggestions and now cite many of them (examples in our responses to the specific comments above).

Reviewer #2

22. In the method section, the detailed derivation of Eq. (2), the main equation of the model, is omitted, which is OK, because the interested readers can read the previous paper for the details. But I would suggest that the authors give some intuitive explanations here with words for better understanding.

Done (main text lines 233-235); we now have added a new Auxiliary Supplementary file (Appendix 2) to provide details on the model's derivation and code to reproduce the results of its numerical evaluations.

23. Eq. 3 is equally important. Can the authors give an example to illustrate how body size is included in the constant B_0 ?

A related question: I don't quite understand how the authors obtained the scaling powers of body mass to plot Figure 1 and 2 in the supplementary materials.

We now explain this in the main text lines 324-332 (Methods) and in SM Section 1.1.

24. The most important and interesting prediction of this study is the hierarchy of the influence of five traits on thermal adaptation, and therefore, my biggest question is: how do we know that this particular order is not the outcome of this particular model and particular parameter chosen for the analysis (such as Fig. 1-C). How do we know this is a general pattern, instead of model-specific? Yes, the data shows it, but how do we prove its universality theoretically? This is a theoretical question, but I haven't figured out the solution. I would like to hear the authors' insight.

The authors did state: "We note that our results do not rely on the specific form or underlying thermodynamic assumptions of the Sharpe-Schoolfield equation..." But they didn't give detailed explanation or proof, so I am not satisfied.

16

The reviewer raises an important philosophical point: *How do we know that a particular mathematical model is the best one to explain an empirical pattern?* We have now qualified further our assumptions and how general this result and our theory is in lines 208-210. We have also now added another Auxiliary Appendix 2 as part of the SM where we explain why this model is generally applicable to arthropods, and cite it in the Methods (lines 233-235).

25. A related question: within the framework of this model, what makes a certain trait, say, development time, more important than the others? Is it due to some specific values of some parameters? Or is it due to their role in Eq. (2)? Or...? I feel that the authors should give some qualitative and intuitive explanations for the biological meanings behind it, (instead of just saying this is the result of our model).

Regarding why development time is the most important trait in terms of its contributions to fitness: we have addressed this more directly now following the comments from Reviewer 1 (see response to their comment 6). More generally, we now explain why the r_m approximation we have derived faithfully captures the qualitative contributions of different traits to it consistent with general life-history theory in lines 208-210, and added an Auxiliary Appendix 2 as part of the SM where we explain why this model is generally applicable to arthropods, and cite it in the Methods (lines 233-235).

26. 5. A minor suggestion: The legend and the text around Fig. 2 is somewhat hard to read. It took me a while to figure out what the authors were trying to say. Maybe it is just me, but maybe the authors can revise it somewhat?

We have now revised the text legend of Fig 2 to address this.

27. 6. Here is another important question: the only reason given by the authors for the potential trade-offs between the Tpk's of life history traits is the fixed energy budget. As an animal energetics guy, I accept this reason. But I feel this might not be enough; there might be some other reasons. For example, the authors mentioned the tradeoffs between development rate and mortality. How would the energy budget argument explain mortality? When it comes to aging, energy budget plays a key role, but when it comes to insects, other external mortality factors, such as predation, instead of intrinsic aging, are more important, so how does energy budget play a role?.

This is a great point. At the level this analysis was run, we wouldn't expect environmental factors, such as predation to be driving patterns across such a broad diversity of arthropods. We have edited the text on lines 142-143 to reflect the possibility of other factors playing a significant role.

Reviewer #3

28. The authors note that their analysis does not consider positive or negative covariation among the life-history traits analyzed (L 242-243). This is understandable given the already complex nature of their analysis. However, trait covariation could alter some of the authors' conclusions, in particular the hierarchy of effects that they have identified on the thermal response of fitness (L 56-72). For example, if developmental time is largely an evolutionary response to juvenile and adult mortality schedules, then saying that developmental time has more of an effect on the thermal response of fitness than do either juvenile or adult mortality may actually not be true. The effects of juvenile & adult mortality may operate indirectly through their effects on developmental time. Whether the authors agree with me or not on this point, I suggest that it might be helpful to discuss the above limitation of their analysis in the Discussion section.

This is a key point, also raised by Reviewer 1; please see our response to their comment 16 above.

29. 2) Are the thermal responses discussed in this paper genotypic evolutionary or phenotypic plastic responses – i.e., 'thermal adaptation' or thermal acclimation' (or both)? The term "thermal adaptation" is frequently used in this manuscript, and the authors focus on temperature effects on evolutionary fitness, so it seems that the authors are talking about evolutionary responses. Please clarify. The answer to this question bears on my 1st comment above.

We agree that this distinction needed to be made clearer so we have revised the text to address this (lines 32-36).

30. L 30: What is a T_{pk} ? Please define at first use. How is it different from T_{opt} , which seems to be the same? Reference here to Fig. 1B is not completely helpful in this respect – it would be helpful to the reader to explain the difference in words in the text (at first use, and not until L 218-220). A reference to the glossary in Table 1 in the Methods would also be helpful.

Clarified on lines 33-34. Also, we have double-checked and revised any confusion in the notation for this throughout.

31. L 153-154: I agree that the relative effects of genotypic evolution and phenotypic plasticity on thermal responses (including the temperature-size rule) should be distinguished. Many investigators have studied the T-size rule with acute T changes in the laboratory. A common explanation for the T-size rule in these cases is that maturation rate is more T-sensitive than growth rate. The authors discuss how evolutionary thermal responses may not follow the T-size rule, but they have not estimated thermal responses of growth rate, which are critical for understanding this rule.

18

We now clarify this issue further in lines 164-170 and mention the need for population growth rate data on lines 174-176.

32. L 200: Change "traits" to "trait"?

Done (line 201).

Reviewer #3

33. I was asked to look at the part of phylogenetic comparative analyses (PCA) in this MS. So I am just going to do that although I did scan the other parts and find them very interesting. I believe the part for PCAs were done well and competently. I have just a minor comments.

1. it says "a Cauchy prior for the random effects covariance matrix,". The parameter expanded prior (called by Hadfield) is a prior using the non-central F distribution so it is not quite a Cauchy although a special case of it is a half Cauchy. So the authors may call it just the parameter expanded prior.

The Reviewer is right. We apologise for the confusion. We now refer to the prior as "parameter expanded" (line 379).

34. 2. I checked the code provided and it is very nice - it could do with more annotation (but it is fine). Actually, they use a "meta-analytic" model incorporating SE^2 (using the mev argument). This could be mentioned in the method section. Actually, in a meta-analytic literature, they recommend modelling covariance between errors (i.e SE^2). Otherwise, you may have more Type 1 error but there is no easy solutions for this (at least one cannot do it with MCMCglmm). For example, see

Mavridis D, Salanti G. A practical introduction to multivariate meta-analysis. *Statistical methods in medical research*. 2013 Apr;22(2):133-58.

3. the authors do not justify why they used 100 trees. Actually, this paper shows 50 trees are enough to correct for phylogenetic uncertainty - so 100 is a good one.

Nakagawa S, De Villemereuil P. A general method for simultaneously accounting for phylogenetic and species sampling uncertainty via Rubin's rules in comparative analysis. *Systematic Biology*. 2019 Jul 1;68(4):632-41.

We thank the Reviewer for bringing this paper to our attention. We estimated 100 different trees to incorporate the phylogenetic uncertainty into our estimates as best as possible. We now make this clearer and cite the suggested paper to justify our use of 100 trees (lines 353-357).

19

References

1. Kingsolver, J. G. & Woods, H. A. Beyond Thermal Performance Curves: Modeling Time-Dependent Effects of Thermal Stress on Ectotherm Growth Rates. *The American Naturalist* **187**, 283–294 (2016).
2. Bernhardt, J. R., Sunday, J. M., Thompson, P. L. & O'Connor, M. I. Nonlinear averaging of thermal experience predicts population growth rates in a thermally variable environment. *Proc. R. Soc. B Biol. Sci.* **285**, 20181076 (2018).
3. Huxley, P. J., Murray, K. A., Pawar, S. & Cator, L. J. The effect of resource limitation on the temperature dependence of mosquito population fitness. *Proc. R. Soc. B* **288**, rspb.2020.3217 (2021).
4. Huxley, P. J., Murray, K. A., Pawar, S. & Cator, L. J. Competition and resource depletion shape the thermal response of population fitness in *Aedes aegypti*. *Commun. Biol* **5**, 1–11 (2022).
5. Flatt, T. Life-History Evolution and the Genetics of Fitness Components in *Drosophila melanogaster*. *Genetics* **214**, 3–48 (2020).
6. Pawar, S., Dell, A. I., Savage, V. M. & Knies, J. L. Real versus Artificial Variation in the Thermal Sensitivity of Biological Traits. *Am. Nat.* **187**, E41–E52 (2016).
7. Stavrinides, M. C. & Mills, N. J. Influence of temperature on the reproductive and demographic parameters of two spider mite pests of vineyards and their natural predator. *BioControl* **56**, 315–325 (2011).
8. Amarasekare, P. & Savage, V. A framework for elucidating the temperature dependence of fitness. *Am. Nat.* **179**, 178–91 (2012).
9. Kontopoulos, D.-G. *et al.* Phytoplankton thermal responses adapt in the absence of hard thermodynamic constraints. *Evolution* **74**, 775–790 (2020).
10. Huey, R. Evolutionary physiology of insect thermal adaptation to cold environments. *Low temperature biology of insects* 223–241 (2010).
11. Hoffmann, A. A. & Sgrò, C. M. Climate change and evolutionary adaptation. *Nature* **470**, 479–85 (2011).
12. Hoffmann, A. A. & Ross, P. A. Rates and Patterns of Laboratory Adaptation in (Mostly) Insects. *Journal of Economic Entomology* **111**, 501–509 (2018).

Imperial College
London

13. Padfield, D., O'Sullivan, H. & Pawar, S. rTPC and nls. multstart: a new pipeline to fit thermal performance curves in R. *Methods in Ecology and Evolution* **12**, 1138–1143 (2021).

Decision Letter, first revision:

13th October 2023

Dear Paul,

Thank you for submitting your revised manuscript "Variation in temperature of peak trait performance constrains adaptation of arthropod populations to climatic warming" (NATECOLEVOL-23051150A). It has now been seen again by the original reviewers and their comments are below. As you will see from their comments, the reviewers find that the paper has greatly improved in revision. Therefore, we'll be happy in principle to publish it in Nature Ecology & Evolution, pending minor revisions to satisfy the reviewers' final requests and to comply with our editorial and formatting guidelines.

Since the reviewers' final requests are relatively minor, I would request you to address their points in a final revised version of your manuscript. Please also prepare a point-by-point response to help us assess your responses.

If your manuscript is in a .pdf format, please email us a copy of the file in an editable format (Microsoft Word or LaTeX)-- we can not proceed with PDFs at this stage.

We will now perform detailed checks on your paper and will send you a checklist detailing our editorial and formatting requirements in about a week. Please do not upload the final materials until you receive this additional information from us.

[REDACTED]

Reviewer #1 (Remarks to the Author):

I like this paper even more this time. It is a gem. And your responses to suggestions were well done, too. I really like this paper. The theme is timely and important, and the approach developed here is insightful.

I have just a few minor comments.

line 112 the ordering of T_p currently goes from highest to lowest. I suggest reversing this so it goes $T_z < T_{zj} < T_{bmax} < T_{alpha}$. Thus low to high, as is "natural" ordering of temperatures.

Fig 3 add units of T_{pk} and E ?

127 us the "evidence" significance?

22127-132. You lost me. If Talpha is closely related to Topt, I'd expect that Talpha would evolve quickly. I must be confused here.

136 This is fantastic. I've brooded about why there might be multiple optimal temperatures for decades. Here's what I wrote in 1982. The passage I wrote is not relevant to your arguments, as I was thinking about Topt for performance traits (speed, digestion, hearing), not life history ones. (Note: I was working with reptiles, and I saw no way to measure the thermal dependence of life history traits in reptiles. So I focused only on performance traits, which I could easily measure. In any case, I really like the way you are treating this related issue.

"Acceptance of the hypothesis of multiple physiological optimal temperatures will lead to two interesting questions: (1) why has selection favored multiple optimal temperatures? (2) why does one system have a higher and another system a lower thermal optimum? Multiple optima seem inefficient because no single body temperature simultaneously optimizes all systems (Huey and Stevenson, 1979). Perhaps the optimal temperatures are related to the thermal conditions at the time (Brett, 1971; Dawson, 1975) or place (Regal, 1980) where the particular system functions."

I just reread Huey et al. 1991. We looked at Partridge's flies that have been evolving for several years at 18 or at 25 ° and measured dev time at both temperatures. The key result: "For example, low-temperature flies developed about 1/2 day faster than did high-temperature flies at 16.5°C. The converse was true at 25°C, but the time difference was only a few hours (Table 1)."

So at least in this case, thermal sensitivity of development time evolved quickly

Huey, R. B., Partridge, L., & Fowler, K. (1991). Thermal sensitivity of *Drosophila melanogaster* responds rapidly to laboratory natural selection. *Evolution*, 45, 751-756.

Note: Linda had a follow up paper: Partridge, L., Barrie, B., Barton, N. H., Fowler, K., & French, V. (1995). Rapid laboratory evolution of adult life history traits in *Drosophila melanogaster* in response to temperature. *Evolution*, 49, 538-544.

226 isn't bx f offspring produced by a *female* of age x?

242 "metabolic trait" ???

255 Something for the future consideration. Your estimates of Topt are based on fixed temperatures. Martin and Huey showed that actual Topt shifts to a lower temperature when there's variance in Tb experienced, especially if the TPC is strongly skewed. Jensen's Inequality once again!

More on fixed temperatures. In my last review, I noted that the lab data come from studies holding critters at fixed temperatures from egg to death. Consider a trait at high temperature. Initially, performance might be high, but continued exposure to that temperature will cause damage and a drop in performance (as per Kingsolver and Woods). That's why I wrote that performance or fitness

23estimates at high temperature - when the exposure is long, as in development time -- will underestimate fitness. My wording was perhaps confusing. I'm guessing that if one exposed the critter to high temperature for only say 12 h, and then exposed them to a lower temperature for 12 h, then one might see a higher trait value than if the critter was held at high temperature for 24 h/day X multiple days. But I need to check the Bernhardt et al. study you cited! I may be wrong. I do know that lizards run fast at high Tb. But if you hold them at that temperature for very long, they can't take it. Hence an "average" of speed over time would drop, and underestimate the maximal speed early on.

I agree about resource causing a left shift - this was demonstrated beautifully by J. R. Brett (1971) with salmonids, and Kingsolver and I did a simple model of this and proposed "metabolic meltdown".

I think we all agree that these relationships are a moving target!

Again, a superb job.

Ray Huey

Note: check capitalization of titles in Lit Cited. ref 12, for example, has first letters capitalized throughout the title.

The citation for this chapter should be:

Huey, R. B. (2010). Evolutionary physiology of insect thermal adaptation to cold environments. In D. L. Denlinger & R. E. Lee, Jr. (Eds.), *Low Temperature Biology of Insects* (pp. 223-241). Cambridge University Press.

Reviewer #2 (Remarks to the Author):

The authors did a great job in this revision. They addressed pretty much all of my questions well. Their answer to my first question regarding Eq. 2 is not satisfactory though (because they didn't really give a qualitative explanation of it in the main text, and one still needs to read the supplementary material), but this is just my own opinion, and it doesn't affect the quality of this paper. Also, I am not satisfied with their answer to my last question regarding the tradeoff. Their answer is too brief, and I expected to see some details. But, again, this is just my opinion, and giving too much detail is probably beyond the scope of this paper. Overall, this is a great paper, and should be published in *Nature Ecology and Evolution*.

Reviewer #3 (Remarks to the Author):

General comments:

In general, I am largely satisfied with the authors' responses to my comments. However, I do not see how the authors' response to point 16 of Reviewer 1 answers my query (point 28 of reviewer 3)

24repeated here:

28. The authors note that their analysis does not consider positive or negative covariation among the life-history traits analyzed (L 242-243). This is understandable given the already complex nature of their analysis. However, trait covariation could alter some of the authors' conclusions, in particular the hierarchy of effects that they have identified on the thermal response of fitness (L 56-72). For example, if developmental time is largely an evolutionary response to juvenile and adult mortality schedules, then saying that developmental time has more of an effect on the thermal response of fitness than do either juvenile or adult mortality may actually not be true. The effects of juvenile & adult mortality may operate indirectly through their effects on developmental time. Whether the authors agree with me or not on this point, I suggest that it might helpful to discuss the above limitation of their analysis in the Discussion section.

Authors' response: This is a key point, also raised by Reviewer 1; please see our response to their comment 16 above.

Specific comments:

L 9: Please insert "of" between "performance" and "four".

Reviewer #4 (Remarks to the Author):

It seems like the authors have addressed the majority of my and others' comments. I note that my Point 2 is not addressed (probably they missed it?)

Our ref: NATECOLEVOL-23051150A

6th November 2023

Dear Dr. Huxley,

Thank you for your patience as we've prepared the guidelines for final submission of your Nature Ecology & Evolution manuscript, "Variation in temperature of peak trait performance constrains adaptation of arthropod populations to climatic warming" (NATECOLEVOL-23051150A). Please carefully follow the step-by-step instructions provided in the attached file, and add a response in each

25row of the table to indicate the changes that you have made. Please also check and comment on any additional marked-up edits we have proposed within the text. Ensuring that each point is addressed will help to ensure that your revised manuscript can be swiftly handed over to our production team.

****We would like to start working on your revised paper, with all of the requested files and forms, as soon as possible (preferably within two weeks). Please get in contact with us immediately if you anticipate it taking more than two weeks to submit these revised files.****

In recognition of the time and expertise our reviewers provide to Nature Ecology & Evolution's editorial process, we would like to formally acknowledge their contribution to the external peer review of your manuscript entitled "Variation in temperature of peak trait performance constrains adaptation of arthropod populations to climatic warming". For those reviewers who give their assent, we will be publishing their names alongside the published article.

Nature Ecology & Evolution offers a Transparent Peer Review option for new original research manuscripts submitted after December 1st, 2019. As part of this initiative, we encourage our authors to support increased transparency into the peer review process by agreeing to have the reviewer comments, author rebuttal letters, and editorial decision letters published as a Supplementary item. When you submit your final files please clearly state in your cover letter whether or not you would like to participate in this initiative. Please note that failure to state your preference will result in delays in accepting your manuscript for publication.

Cover suggestions

We welcome submissions of artwork for consideration for our cover. For more information, please see our https://www.nature.com/documents/Nature_covers_author_guide.pdf target="new"> guide for cover artwork.

Nature Ecology & Evolution has now transitioned to a unified Rights Collection system which will allow our Author Services team to quickly and easily collect the rights and permissions required to publish

26your work. Approximately 10 days after your paper is formally accepted, you will receive an email in providing you with a link to complete the grant of rights. If your paper is eligible for Open Access, our Author Services team will also be in touch regarding any additional information that may be required to arrange payment for your article.

Please note that *Nature Ecology & Evolution* is a Transformative Journal (TJ). Authors may publish their research with us through the traditional subscription access route or make their paper immediately open access through payment of an article-processing charge (APC). Authors will not be required to make a final decision about access to their article until it has been accepted. [Find out more about Transformative Journals](https://www.springernature.com/gp/open-research/transformative-journals)

Authors may need to take specific actions to achieve [compliance with funder and institutional open access mandates](https://www.springernature.com/gp/open-research/funding/policy-compliance-faqs). If your research is supported by a funder that requires immediate open access (e.g. according to [Plan S principles](https://www.springernature.com/gp/open-research/plan-s-compliance)) then you should select the gold OA route, and we will direct you to the compliant route where possible. For authors selecting the subscription publication route, the journal's standard licensing terms will need to be accepted, including [self-archiving-and-license-to-publish](https://www.nature.com/nature-portfolio/editorial-policies/self-archiving-and-license-to-publish). Those licensing terms will supersede any other terms that the author or any third party may assert apply to any version of the manuscript.

[REDACTED]

[REDACTED]

Reviewer #1:

Remarks to the Author:

I like this paper even more this time. It is a gem. And your responses to suggestions were well done, too. I really like this paper. The theme is timely and important, and the approach developed here is insightful.

27I have just a few minor comments.

line 112 the ordering of T_p currently goes from highest to lowest. I suggest reversing this so it goes $T_z < T_{zj} < T_{bmax} < T_{alpha}$. Thus low to high, as is "natural" ordering of temperatures.

Fig 3 add units of T_{pk} and E ?

127 us the "evidence" significance?

127-132. You lost me. If T_{alpha} is closely related to T_{opt} , I'd expect that T_{alpha} would evolve quickly. I must be confused here.

136 This is fantastic. I've brooded about why there might be multiple optimal temperatures for decades. Here's what I wrote in 1982. The passage I wrote is not relevant to your arguments, as I was thinking about T_{opt} for performance traits (speed, digestion, hearing), not life history ones. (Note: I was working with reptiles, and I saw no way to measure the thermal dependence of life history traits in reptiles. So I focused only on performance traits, which I could easily measure. In any case, I really like the way you are treating this related issue.

"Acceptance of the hypothesis of multiple physiological optimal temperatures will lead to two interesting questions: (1) why has selection favored multiple optimal temperatures? (2) why does one system have a higher and another system a lower thermal optimum? Multiple optima seem inefficient because no single body temperature simultaneously optimizes all systems (Huey and Stevenson, 1979). Perhaps the optimal temperatures are related to the thermal conditions at the time (Brett, 1971; Dawson, 1975) or place (Regal, 1980) where the particular system functions."

I just reread Huey et al. 1991. We looked at Partridge's flies that have been evolving for several years at 18 ° or at 25 ° and measured dev time at both temperatures. The key result: "For example, low-temperature flies developed about 1/2 day faster than did high-temperature flies at 16.5°C. The converse was true at 25°C, but the time difference was only a few hours (Table 1)."

So at least in this case, thermal sensitivity of development time evolved quickly

Huey, R. B., Partridge, L., & Fowler, K. (1991). Thermal sensitivity of *Drosophila melanogaster* responds rapidly to laboratory natural selection. *Evolution*, 45, 751-756.

Note: Linda had a follow up paper: Partridge, L., Barrie, B., Barton, N. H., Fowler, K., & French, V. (1995). Rapid laboratory evolution of adult life history traits in *Drosophila melanogaster* in response to temperature. *Evolution*, 49, 538-544.

226 isn't bx f offspring produced by a *female* of age x?

242 "metabolic trait" ???

255 Something for the future consideration. Your estimates of T_{opt} are based on fixed temperatures. Martin and Huey showed that actual T_{opt} shifts to a lower temperature when there's variance in T_b experienced, especially if the TPC is strongly skewed. Jensen's Inequality once again!

More on fixed temperatures. In my last review, I noted that the lab data come from studies holding critters at fixed temperatures from egg to death. Consider a trait at high temperature. Initially, performance might be high, but continued exposure to that temperature will cause damage and a drop in performance (as per Kingsolver and Woods). That's why I wrote that performance or fitness estimates at high temperature - when the exposure is long, as in development time -- will underestimate fitness. My wording was perhaps confusing. I'm guessing that if one exposed the critter to high temperature for only say 12 h, and then exposed them to a lower temperature for 12 h, then one might see a higher trait value than if the critter was held at high temperature for 24 h/day X multiple days. But I need to check the Bernhardt et al. study you cited! I may be wrong. I do know that lizards run fast at high T_b . But if you hold them at that temperature for very long, they can't take it. Hence an "average" of speed over time would drop, and underestimate the maximal speed early on.

I agree about resource causing a left shift - this was demonstrated beautifully by J. R. Brett (1971) with salmonids, and Kingsolver and I did a simple model of this and proposed "metabolic meltdown".

I think we all agree that these relationships are a moving target!

Again, a superb job.

Ray Huey

Note: check capitalization of titles in Lit Cited. ref 12, for example, has first letters capitalized throughout the title.

The citation for this chapter should be:

Huey, R. B. (2010). Evolutionary physiology of insect thermal adaptation to cold environments. In D. L. Denlinger & R. E. Lee, Jr. (Eds.), *Low Temperature Biology of Insects* (pp. 223-241). Cambridge University Press.

Reviewer #2:

Remarks to the Author:

The authors did a great job in this revision. They addressed pretty much all of my questions well. Their answer to my first question regarding Eq. 2 is not satisfactory though (because they didn't really give a qualitative explanation of it in the main text, and one still needs to read the supplementary material), but this is just my own opinion, and it doesn't affect the quality of this paper. Also, I am not satisfied with their answer to my last question regarding the tradeoff. Their answer is too brief, and I expected to see some details. But, again, this is just my opinion, and giving too much detail is

29probably beyond the scope of this paper. Overall, this is a great paper, and should be published in Nature Ecology and Evolution.

Reviewer #3:
Remarks to the Author:
General comments:

In general, I am largely satisfied with the authors' responses to my comments. However, I do not see how the authors' response to point 16 of Reviewer 1 answers my query (point 28 of reviewer 3) repeated here:

28. The authors note that their analysis does not consider positive or negative covariation among the life-history traits analyzed (L 242-243). This is understandable given the already complex nature of their analysis. However, trait covariation could alter some of the authors' conclusions, in particular the hierarchy of effects that they have identified on the thermal response of fitness (L 56-72). For example, if developmental time is largely an evolutionary response to juvenile and adult mortality schedules, then saying that developmental time has more of an effect on the thermal response of fitness than do either juvenile or adult mortality may actually not be true. The effects of juvenile & adult mortality may operate indirectly through their effects on developmental time. Whether the authors agree with me or not on this point, I suggest that it might helpful to discuss the above limitation of their analysis in the Discussion section.

Authors' response: This is a key point, also raised by Reviewer 1; please see our response to their comment 16 above.

Specific comments:

L 9: Please insert "of" between "performance" and "four".

Reviewer #4:
Remarks to the Author:
It seems like the authors have addressed the majority of my and others' comments. I note that my Point 2 is not addressed (probably they missed it?)

Author Rebuttal, first revision:Responses to the Reviewers

Comments made by Reviewers are in **blue text**, whereas our responses are in **black text**.

All revisions to the manuscript referenced below are in **red text** in the revised manuscript.

Reviewer #1

1. line 112 the ordering of T_p currently goes from highest to lowest. I suggest reversing this so it goes $T_z < T_{zj} < T_{bmax} < T_{alpha}$. Thus low to high, as is “natural” ordering of temperatures.

The ordering from *greatest to smallest impacts* on thermal fitness directly is a key finding of this study, so we prefer to keep it as such.

2. Fig 3 add units of T_{pk} and E ?

Done.

3. 127 is the “evidence” significance?

This result is indeed not statistically significant given that the 95% Highest Posterior Density intervals of the evolutionary rates of T_{pk}^{α} and $T_{pk}^{b,max}$ overlap (Fig. 4C). To avoid misinterpretation, we have now qualified this result (line 126).

4. 127-132. You lost me. If T_{alpha} is closely related to T_{opt} , I'd expect that T_{alpha} would evolve quickly. I must be confused here.

Our result is about stabilising selection over long “macroevolutionary” timescales (during which new species are formed) whereas the reviewer is thinking about evolution of short “microevolutionary” timescales. As such low evolutionary rate over long timescales due to stabilizing selection and directional changes over short timescales under environmental change that the Reviewer is alluding to go hand-in-hand. We have further clarified this in revised lines 126-134 of the main text (also see our response to comment #5 below).

5. I just reread Huey et al. 1991. We looked a Partridge's flies that have been evolving for several years at 18 or at 25 ° C and measured dev time at both temperatures. The key result: “For example, low- temperature flies developed about 1/2 day faster than did high-temperature flies at 16.5° C. The converse was true at 25” C, but the time difference was only a few hours (Table 1).” So at least in this case, thermal sensitivity of development time evolved quickly

Huey, R. B., Partridge, L., & Fowler, K. (1991). Thermal sensitivity of *Drosophila melanogaster* responds rapidly to laboratory natural selection. *Evolution*, 45, 751-756.

32

Note: Linda had a follow up paper: Partridge, L., Barrie, B., Barton, N. H., Fowler, K., & French, V. (1995). Rapid laboratory evolution of adult life history traits in *Drosophila melanogaster* in response to temperature. *Evolution*, 49, 538-544.

These studies provide evidence that development rate can evolve in response to a new temperature regime, but they do not quantify thermal performance at these temperatures (thus we do not know if T_{pk} alpha has shifted) or whether these short-term changes are faster for α relative to other traits. However, we agree that the issue of macro- vs micro-evolutionary changes in T_{pk} s need to be clarified, and we have now done so in lines 126-134 of the revised main text, where we now also cite the Partridge et al 1995 paper.

6. 226 isn't bx f offspring produced by a *female* of age x?

Done.

7. 242 "metabolic trait" ???

We have now qualified this more precisely: lines 241-242.

8. 255 Something for the future consideration. Your estimates of T_{opt} are based on fixed temperatures. Martin and Huey showed that actual T_{opt} shifts to a lower temperature when there's variance in T_b experienced, especially if the TPC is strongly skewed. Jensen's Inequality once again! ... More on fixed temperatures. In my last review, I noted that the lab data come from studies holding critters at fixed temperatures from egg to death. Consider a trait at high temperature. Initially, performance might be high, but continued exposure to that temperature will cause damage and a drop in performance (as per Kingsolver and Woods). That's why I wrote that performance or fitness estimates at high temperature - when the exposure is long, as in development time - will underestimate fitness. My wording was perhaps confusing. I'm guessing that if one exposed the critter to high temperature for only say 12 h, and then exposed them to a lower temperature for 12 h, then one might see a higher trait value than if the critter was held at high temperature for 24 h/day X multiple days. But I need to check the Bernhardt et al. study you cited! I may be wrong. I do know that lizards run fast at high T_b . But if you hold them at that temperature for very long, they can't take it. Hence an "average" of speed over time would drop, and underestimate the maximal speed early on.

Agreed. We have addressed these points in greater detail in lines 176-184.

9. Note: check capitalization of titles in Lit Cited. ref 12, for example, has first letters capitalized throughout the title.

The citation for this chapter should be: Huey, R. B. (2010). Evolutionary physiology of insect thermal adaptation to cold environments. In D. L. Denlinger & R. E. Lee, Jr. (Eds.), *Low Temperature Biology of Insects* (pp. 223-241). Cambridge University Press.

Thank you for pointing this out. We have now made sure that all of our citations and references conform to the Nature style.

Reviewer #2

10. Their answer to my first question regarding Eq. 2 is not satisfactory though (because they didn't really give a qualitative explanation of it in the main text, and one still needs to read the supplementary material), but this is just my own opinion, and it doesn't affect the quality of this paper.

We appreciate the reviewer's point of view on this, but prefer to keep these details in the supplementary material to make sure we don't overwhelm a less technically-inclined reader.

11. Also, I am not satisfied with their answer to my last question regarding the tradeoff. Their answer is too brief, and I expected to see some details. But, again, this is just my opinion, and giving too much detail is probably beyond the scope of this paper.

We agree with the reviewer that this is an interesting question that is beyond the scope of this paper. We appreciate the point and believe that better understanding the mechanisms of these trade-offs (apart from energy budget) is an important area of future work for the field.

Reviewer #3

12. In general, I am largely satisfied with the authors' responses to my comments. However, I do not see how the authors' response to point 16 of Reviewer 1 answers my query (point 28 of reviewer 3) repeated here:

28. The authors note that their analysis does not consider positive or negative covariation among the life-history traits analyzed (L 242-243). This is understandable given the already complex nature of their analysis. However, trait covariation could alter some of the authors' conclusions, in particular the hierarchy of effects that they have identified on the thermal response of fitness (L 56-72). For example, if developmental time is largely an evolutionary response to juvenile and adult mortality schedules, then saying that developmental time has more of an effect on the thermal response of fitness than do either juvenile or adult mortality may actually not be true. The effects of juvenile & adult mortality may operate indirectly through their effects on developmental time. Whether the authors agree with me or not on this point, I suggest that it might helpful to discuss the above limitation of their analysis in the Discussion section.

34

Authors' response: This is a key point, also raised by Reviewer 1; please see our response to their comment 16 above.

We apologise for overlooking this point. We have addressed this on lines 91-94 of the revised main text. Also note that our phylogenetic analysis reveals predominantly positive correlations between trait T_{pkS} , as discussed in lines 152-160.

13. L 9: Please insert "of" between "performance" and "four".

Done.

Reviewer #4

14. It seems like the authors have addressed the majority of my and others' comments. I note that my Point 2 is not addressed (probably they missed it?):

2. I checked the code provided and it is very nice - it could do with more annotation (but it is fine). Actually, they use a "meta-analytic" model incorporating SE^2 (using the mev argument). This could be mentioned in the method section. Actually, in a meta-analytic literature, they recommend modelling covariance between errors (i.e SE^2). Otherwise, you may have more Type 1 error but there is no easy solutions for this (at least one cannot do it with MCMCglmm). For example, see Mavridis D, Salanti G. A practical introduction to multivariate meta-analysis. *Statistical methods in medical research.* 2013 Apr;22(2):133-58.

Thanks for pointing this out. As you mention, there aren't any easy solutions to this. As the Reviewer suggests, we have clarified this in the Methods (lines 379-381).

Final Decision Letter:

7th December 2023

Dear Dr Huxley,

We are pleased to inform you that your Article entitled "Variation in temperature of peak trait performance constrains adaptation of arthropod populations to climatic warming", has now been accepted for publication in Nature Ecology & Evolution.

Over the next few weeks, your paper will be copyedited to ensure that it conforms to Nature Ecology and Evolution style. Once your paper is typeset, you will receive an email with a link to choose the appropriate publishing options for your paper and our Author Services team will be in touch regarding any additional information that may be required

Due to the importance of these deadlines, we ask you please us know now whether you will be difficult to contact over the next month. If this is the case, we ask you provide us with the contact information (email, phone and fax) of someone who will be able to check the proofs on your behalf, and who will be available to address any last-minute problems . Once your paper has been scheduled for online publication, the Nature press office will be in touch to confirm the details.

Acceptance of your manuscript is conditional on all authors' agreement with our publication policies (see www.nature.com/authors/policies/index.html). In particular your manuscript must not be published elsewhere and there must be no announcement of the work to any media outlet until the publication date (the day on which it is uploaded onto our web site).

Please note that *Nature Ecology & Evolution* is a Transformative Journal (TJ). Authors may publish their research with us through the traditional subscription access route or make their paper immediately open access through payment of an article-processing charge (APC). Authors will not be required to make a final decision about access to their article until it has been accepted. [Find out more about Transformative Journals](https://www.springernature.com/gp/open-research/transformative-journals)

Authors may need to take specific actions to achieve [compliance with funder and institutional open access mandates](https://www.springernature.com/gp/open-research/funding/policy-compliance-faqs). If your research is supported by a funder that requires immediate open access (e.g. according to [Plan S principles](https://www.springernature.com/gp/open-research/plan-s-compliance)) then you should select the gold OA route, and we will direct you to the compliant route where

36possible. For authors selecting the subscription publication route, the journal's standard licensing terms will need to be accepted, including <https://www.nature.com/nature-portfolio/editorial-policies/self-archiving-and-license-to-publish>. Those licensing terms will supersede any other terms that the author or any third party may assert apply to any version of the manuscript.

We welcome the submission of potential cover material (including a short caption of around 40 words) related to your manuscript; suggestions should be sent to Nature Ecology & Evolution as electronic files (the image should be 300 dpi at 210 x 297 mm in either TIFF or JPEG format). Please note that such pictures should be selected more for their aesthetic appeal than for their scientific content, and that colour images work better than black and white or grayscale images. Please do not try to design a cover with the Nature Ecology & Evolution logo etc., and please do not submit composites of images related to your work. I am sure you will understand that we cannot make any promise as to whether any of your suggestions might be selected for the cover of the journal.

You can generate the link yourself when you receive your article DOI by entering it here: <http://authors.springernature.com/share>.

Yours sincerely,

[REDACTED]

P.S. Click on the following link if you would like to recommend Nature Ecology & Evolution to your

37librarian <http://www.nature.com/subscriptions/recommend.html#forms>

** Visit the Springer Nature Editorial and Publishing website at http://editorial-jobs.springernature.com?utm_source=ejp_NEcoE_email&utm_medium=ejp_NEcoE_email&utm_campaign=ejp_NEcoE for more information about our career opportunities. If you have any questions please click [here](mailto:editorial.publishing.jobs@springernature.com).**